



# MLAir (v1.0) - a tool to enable fast and flexible machine learning on air data time series

Lukas H. Leufen[1,2], Felix Kleinert[1,2], and Martin G. Schultz[1]

[1]Research Centre Jülich, Jülich Supercomputing Centre, Germany
[2]University of Bonn, Institute of Geosciences, Germany

**Correspondence:** LH Leufen (l.leufen@fz-juelich.de)

**Abstract.** With MLAir (Machine Learning on Air data) we created a software environment that simplifies and accelerates the exploration of new machine learning (ML) models for the analysis and forecasting of meteorological and air quality time series. Thereby MLAir is not developed as an abstract workflow, but hand in hand with actual scientific questions. It thus addresses scientists with either a meteorological or a ML background. Due to their relative ease of use and spectacular results in other

application areas, neural networks and other ML methods are gaining enormous momentum also in the weather and air quality research communities. Even though there are already many books and tutorials describing how to conduct a ML experiment, there are many stumbling blocks for a newcomer. In contrast, people familiar with ML concepts and technology often have difficulties understanding the nature of atmospheric data. With MLAir we have addressed a number of these pitfalls so that it becomes easier for scientists of both domains to rapidly start off their ML application. MLAir has been developed in such a way

that it is easy to use and is designed from the very beginning as a standalone, fully functional experiment. Due to its flexible, modular code base, code modifications are easy and personal experiment schedules can be quickly derived. The package also includes a set of simple validation tools to facilitate the evaluation of ML results using standard meteorological statistics. MLAir can easily be ported onto different computing environments from desktop workstations to high-end supercomputers with or without graphics processing units (GPU).

## 1   Introduction

In times of rising awareness of air quality and climate issues, the investigation of air quality and weather phenomena is moving into high focus. Trace substances such as ozone, nitrogen oxides or particulate matter pose a serious health hazard to humans, animals and nature (Cohen et al., 2005; Bentayeb et al., 2015; World Health Organization, 2013; Lefohn et al., 2018; Mills

et al., 2018; US Environmental Protection Agency, 2020). Accordingly, the analysis and prediction of air quality are of great importance in order to be able to initiate appropriate countermeasures or issue warnings. Likewise, impacts of severe weather can be disastrous leading to losses of lives and great economic damage. Weather prediction has been established operationally





in many countries and has become a multi-million dollar industry, creating and selling specialized data products for many different target groups.

These days, forecasts of weather (as a common generic term for atmospheric chemistry, air quality, and meteorology) are generally made with the help of so-called Eulerian grid point models. This type of models, which solve physical (and chemical) equations, operate on grid structures. In fact, however, local observations of weather and air quality are strongly influenced by the immediate environment. For instance, it is quite difficult for atmospheric chemistry models to represent very small-scale problems due to the limited grid resolution of these models and other limitations. Consequently, both global models

and so-called small-scale models, whose grid resolution is still in the magnitude of about a kilometre and thus rather coarse in comparison to local-scale phenomena in the vicinity of a measurement site, show a high uncertainty of the results (c.f. Vautard, 2012; Brunner et al., 2015). To enhance the model output, approaches focusing on the individual point measurements at weather and air quality monitoring stations through downscaling methods are applied allowing local effects to be taken into account. Unfortunately, these methods, being optimized for specific locations, cannot be generalized for other regions and need

to be re-trained for each measurement site.

In a complementary way to traditional downscaling techniques like linear regression and other statistical methods, the use of machine learning (ML) is a promising approach to predict point observations. Methods such as neural networks are able to recognize and reproduce underlying and complex relationships in data sets. Especially driven by computer vision and speech recognition, technologies like convolutional neural networks (CNN, Lecun et al., 1998) or recurrent networks variations such as

long short term memory (LSTM, Hochreiter and Schmidhuber, 1997) or gated recurrent units (GRU, Cho et al., 2014) but also more advanced concepts like variational autoencoders (VAE, Kingma and Welling, 2014; Rezende et al., 2014), or generative adversarial networks (GAN, Goodfellow et al., 2014) are powerful and widely used successfully.

Although the scientific areas of ML and meteorology exists for many years, combining both disciplines is still a formidable challenge, because scientists from these areas do not speak the same language. Meteorologists are used to build models on

the basis of physical equations and empirical relationships from field experiments, and they evaluate their models with data. In contrast, ML scientists use data to build their models on and evaluate either with additional independent data or physical constraints. This elementary difference can lead to misinterpretation of studies and results so that, for example, the ability of the network to generalize is misjudged. Another frequent problem of published studies on ML approaches to weather fore-casting is an incomplete reporting of ML parameters, hyperparameters and data preparation steps that are key to comprehend

and reproduce the work that was done. As shown by Musgrave et al. (2020) these issues are not limited to meteorological applications of ML only.

To further advance the application of ML in the meteorological area, easily accessible solutions to run and document ML experiments together with readily available and fully documented benchmark data sets are urgently needed (c.f. Schultz et al., 2020, forthcoming). Such solutions need to be understandable by both, the ML and meteorological communities and help both

sides to prevent unconscious blunders. A well-designed workflow embedded in a meteorological and ML related environment while accomplishing subject-specific requirements will bring forward the usage of ML in this specific research area.





In this paper, we present a new framework to enable fast and flexible Machine Learning on Air data time series (MLAir). Fast means that MLAir is distributed as full end-to-end framework and thereby simple to deploy. It also allows to deploy typical optimization techniques in ML workflows, and offers further technical features like the use of graphics processing
units (GPU). Concurrent to a simple usage with low barriers for ML-callow scientists, MLAir also offers high customization potential for advanced ML users and can therefore be employed in real-world applications. For example, more complex model architectures can be easily integrated. ML experts who want to explore weather data will find MLAir helpful as it enforces certain standards of the meteorological community. For example, its data preparation step acknowledges the auto-correlation which is typically seen in meteorological time series, and its validation package reports skill scores, i.e. improvement of the
forecast compared to reference models such as persistence and climatology. From a software design perspective, MLAir has been developed according to state-of-the-art software development practices.

This work is structured as follows. Section 2 introduces MLAir by expounding the general design behind the MLAir workflow and showing three application examples to allow the reader to get a general understanding of the tool. Furthermore, we show how the results of an experiment conducted by MLAir are structured and which statistical analysis is applied. This is fol-
70 lowed by section 3, where the considerations and decisions regarding program design are discussed. Section 4 extends further into the configuration options of an experiment and details on customization. Section 5 delineates the limitations of MLAir and discusses for which applications the tool might not be suitable. Finally, section 6 concludes with an overview and outlook on planned developments for the future. We would like to mention that MLAir is an Open Source project and contributions from all communities are welcome.

## 2 MLAir Workflow

The overall goal of designing MLAir was to create a ready-to-run machine learning application for the task of forecasting weather and air quality time series. The tool should allow many customization options to enable users to easily create a custom ML workflow, while at the same time it should support users in executing ML experiments properly and evaluate their results according to accepted standards of the meteorological community. In this section, we will first introduce the general workflow
of MLAir (section 2.1). Then we show with three short examples how MLAir behaves during an experiment and how first modifications can be made (section 2.2). In section 2.3, we then explain how the output of a MLAir experiment is structured and which graphics are created. Finally, we briefly touch on the statistical part of the model evaluation (section 2.4).

### 2.1 Design of the MLAir Workflow

To achieve as many customization as possible and support users as much as required, MLAir had to be designed as an end-to-
85 end workflow comprising all required steps of the time series forecasting task. The workflow of MLAir is controlled by a *Run Environment*, which provides a central data store, performs logging and ensures the orderly execution of a sequence of individual stages. Different workflows can be defined and executed under the umbrella of this environment. The standard MLAir





workflow contains a sequence of typical steps for ML experiments as indicated by Fig. 1: experiment setup, preprocessing, model setup, training, and postprocessing.

Besides the run environment, the experiment setup plays a very important role. During experiment setup, all customization and configuration modules, like the *Data Handler* (section 3.4), *Model Class* (section 3.3), or hyperparameters, are collected and made available to MLAir. Later in the ongoing workflow, these modules are then queried, e.g. the hyperparameters are used in training whereas the *Data Handler* is responsible for an accurate use of the data and therefore already used in the preprocessing. We want to mention that apart from this default workflow, it is also possible to define completely new stages
and integrate them into a custom MLAir workflow (see section 4.8).

Before we go into details on the design considerations and choices and dive deeper into available features and the actual implementation, we show some basic examples of the MLAir usage to demonstrate the underlying ideas and concepts (section 2.2) and afterwards we comment on the results of an experiment (section 2.3) and the statistical analysis (section 2.4).

### 2.2 Running first experiments with MLAir

To install MLAir, the program can be downloaded as described in the *Code Availability* section and the Python library dependencies should be installed from the requirements file. To test the installation, MLAir can be run in a default configuration with no extra arguments (see Fig. 2). These two commands will execute the workflow depicted in Fig. 1. This will perform an ML forecasting experiment of daily maximum ground-level ozone concentrations using a simple feed-forward neural network based on seven input variables consisting of preceding trace gas concentrations of ozone and nitrogen dioxide, and the values
of temperature, humidity, wind speed, cloud cover, and the planetary boundary layer height.

MLAir uses the *Default Data Handler* (see section 4.4) if not explicitly stated and automatically starts downloading all required air quality and meterological data from the Jülich Open Web Interface (Schultz et al., 2017a, b, JOIN) the first time it is executed after a fresh installation. This web interface provides access to a database of measurements of over 10,000 air quality monitoring stations worldwide. In the default configuration, 21-year time series of nine variables from five stations
are retrieved with a daily aggregated resolution. The retrieved data are stored locally to save time on the next execution (the data extraction can of course be configured as described in section 4.4). It is also possible to replace the *Default Data Handler* with a self-made *Data Handler* to use other data sources or read in different data structures. An introduction to this is given in section 3.4.

After preprocessing this data, splitting them into training, validation, and test data, and converting them to a *xarray* and
115 *NumPy* format (details in section 3.1), MLAir creates a new vanilla feed-forward neural network and starts to train it. Finally, the results are evaluated according to meteorological standards and a default set of plots is created. The trained model, all results and forecasts, the experiment parameters and log files, as well as the default plots are pooled in a folder in the current working directory. Thus, in its default configuration, MLAir performs a meaningful meteorological ML experiment, which can serve as a benchmark for further developments and baseline for more sophisticated ML architectures.
In the second example (Fig. 3), we expand the number of precedent time steps as model inputs to provide more contextual information to the vanilla model. Furthermore, we use a different set of observational stations. Therefore, we need to adjust





both parameters *window_history_size* and *stations* in the run call. From a first glance, the output of the experiment run is quite similar to the earlier example. However, there are a couple of aspects in this second experiment, which we would like to point out. Firstly, the *Default Data Handler* of MLAir keeps track of data available locally and thus reduces the overhead of

reloading data from the web if this is not necessary. Therefore, no new data was downloaded for one of the stations (*DEBW107*), because these data had been stored locally already in our first experiment. Of course the *Default Data Handler* can be forced to reload all data from its source if needed (see section 4.1). The second key aspect to highlight here is that the parameter *window_history_size* could be changed and the network was trained anew without any problem even though this change affects the shape of the input data and thus the neural network architecture. Concerning the network output, the second experiment

overwrites all results from the first run, because without an explicit setting of the file path, MLAir always uses the same sandbox directory called *testrun_network*. In a real-world sequence of experiments, we recommend to always specify a new experiment path with a reasonably descriptive name (details on the experiment path in section 4.1).

The third example in this section demonstrates the activation of a partial workflow, namely a re-evaluation of a previously trained neural network. We want to rerun the evaluation part with a different set of stations to perform an independent validation.

This partial workflow would also be employed if the model is supposed to run in production. As we replace the stations for the new evaluation, we need to create a new testing set, but we want to skip the model creation and training steps. Hence, the parameters *create_new_model* and *train_model* are set to *False* (see Fig. 4). With this setup, the model is loaded from the local file path and the evaluation is performed on the newly provided stations. By combining the stations from the second and third experiment in the station parameter the model can be evaluated at all of these stations together. In this setting, MLAir will fail

to execute the evaluation if parameters pertinent for preprocessing or model compilation changed compared to the training run.

It is also possible to continue training of an already trained model. If the *train_model* parameter is set to *True*, training will be resumed at the last epoch reached, if this epoch number is lower than the final epoch setting. Use cases for this are either an experiment interruption (for example due to wallclock time limit exceedance on batch systems) or the desire to extend the training if the optimal network weights have not been found yet. Further details on training resumption can be found in

section 4.9.

## 2.3 Results of an Experiment

All results of an experiment are stored in the directory, which is defined during the *Experiment Setup* stage (see section 4.1). The sub directory structure is created at the beginning of the experiment. There is no automatic deletion of files in case of aborted runs so that the information that is generated up to the program termination can be inspected to find potential errors or

to check on a successful initialization of the model, etc. Fig. 5 shows the output file structure. The content of each directory is as follows:

– All samples used for training and validation are stored in the *batch_data* folder. Even if the batch data could be used further, they serve rather as auxiliary files.





- *forecasts* contains the actual predictions of the trained model. For comparison, MLAir provides two additional forecasts,
first an ordinary multi-linear least squared fit trained on the same data like the ML model and second a persistence
forecast, where observations of the past represent the forecast for the next steps within the prediction horizon. For daily
data, the persistence forecast refers to the last observation of each sample to hold for all forecast steps. All forecasts
(model and references) are provided in normalized and original value ranges. Additionally, the bootstrap forecasts are
stored here (see section 2.4).

- In *latex_report*, there are publication-ready tables in *MarkDown* or *LaTex* format, which give a summary about the used
stations, the number of samples, and the hyperparameters and experiment settings.

- The *logging* folder contains information about the execution of the experiment. In addition to the console output, MLAir
also stores messages on the debugging level, which give a better understanding of the internal program sequence. MLAir
has a tracking functionality, which can be used to trace which data have been stored and pulled from the central data store.
In combination with the corresponding *tracking* plot that is created at the very end of each experiment automatically,
it allows to visually track which parameters have an effect on which stage. This functionality is most interesting for
developers who make modifications to the source code and want to ensure that their changes don't break the data flow.

- The folder *model* contains everything that is related to the trained model. Beside the file, which contain the model itself
(stored in the binary *Hierarchical Data Format, .h5*), there is also an overview graphic of the model architecture and
all callbacks, for example from the learning rate. If a training is not started from the beginning but is either continued
or applied to a pre-trained model, all necessary information like the model or required callbacks must be stored in this
subfolder.

- The *plots* directory contains all graphics that are created during an experiment. Which graphics are to be created in post-
processing can be determined using the *plot_list* parameter in the *Experiment Setup*. In addition, MLAir automatically
generates monitoring plots for instance of the evolution of the loss during training.

As described in the last bullet, all plots which are created during an MLAir experiment can be found in the subfolder *plots*.
By default, all available plot types are created. By explicitly naming individual graphics in the *plot_list* parameter, it is possible
to override this behaviour and specify which graphics are created during postprocessing. Additional plots are created to monitor
the training behaviour. These graphics are always created when a training session is carried out. Most of the plots which are
created in the course of postprocessing are publication-ready graphics with complete legend and resolution of 500 dpi. If it is
intended to add own graphics in MLAir, these graphics can be added to the workflow by attaching an additional run module
(see section 4.8) including the graphic creation methods.

A general overview of the underlying data can be obtained with the graphics *station map* and *data availability*. The *station
map* (Fig. 6) marks the geographical position of the used stations on a plain map with a land-sea mask, country boundaries and
major waters. The *data availability* chart (Fig. 7) indicates the time periods for which preprocessed data for each measuring





station are available. The index data availability also shows whether a station with measurements is available at all for a point in time. In addition, the three subsets for training, validation and testing are highlighted in different colours.

The monitoring graphics show the course of the loss function as well as the error depending on the epoch for the training and validation data (c.f. Fig. 8). In addition, the error of the best model state with respect to the validation data is shown in the heading. If the learning rate is modified during the course of the experiment, another plot is created to show its development. These monitoring graphics are kept as simple as possible and are meant to give an insight into the training process. The underlying data are always stored in the *json* format in the subfolder *model* and can therefore be used for customized plots.

Through the graphs *monthly summary* and *time series* it is possible to review the forecast of the ML model. The *monthly summary* plot (see Fig. 9) summarizes, according to its name, all predictions of the model covering all stations but considering each month separately as box-and-whisker. With this graph it is possible to get a general overview of the distribution of the predicted values compared to the distribution of the observed values for each month. Besides, the exact course of the time series compared to the observation can be viewed in the *time series* plot (not included as figure). However, since this plot has to scale according to the length of the time series, it should be noted that this last-mentioned graph is kept very simple and rather not suitable for publication.

## 2.4 Statistical analysis of results

A central element of MLAir is the statistical evaluation of the results according to state-of-the-art methods used in meteorology. To obtain specific information on the forecasting model, we treat forecasts and observations as random variables. Therefore, the joint distribution $p(m, o)$ of a model $m$ and an observation $o$ contains information on $p(m)$, $p(o)$ (marginal distribution) and the relation $p(o|m)$, $p(m|o)$ (conditional distribution) between both of them (Murphy and Winkler, 1987). Following Murphy et al. (1989), the marginal distribution is shown as histogram (lite grey), while the conditional distribution is shown as percentiles in different line styles. MLAir automatically creates plots for the entire test period (Fig. 10) and, as is common in meteorology, separated by seasons.

In order to access the genuine added value of a new forecasting model, it is essential to take other existing forecasting models into account instead of reporting only metrics related to the observation. In MLAir we implemented three types of basic reference forecasts; i) a persistence forecast, ii) an ordinary least square model and iii) four climatological forecasts.

The persistence forecast is based on the last observed time step, which is then used as a prediction for all lead times. The ordinary least square model serves as a linear competitor and is derived from the same data the model was trained with. For the climatological references, we follow Murphy (1988) who defined single and multi valued climatological references based on different time scales. We refer the reader to Murphy (1988) for an in-depth discussion on the climatological reference. Note, that this kind of persistence and also the climatological forecast might not be applicable for all temporal resolutions and therefore may need adjustment. We think here, for example, of a clear diurnal pattern in temperature, for which a persistence of successive observations would not provide a good forecast. In this context, a reference forecast based on the observation of the previous day at the same time might be more suitable.



For the comparison, we use a skill score $S$, which is essentially the ratio of the performance of a new forecast and a
220 competitive reference with respect to a statistical metric (e.g. mean squared error). A positive skill score can be interpreted as
the percentage of improvement of the new model forecast in comparison to the reference. On the other hand, a negative skill
score denotes that the forecast of interest is worse than the referencing forecast. Consequently, a value of zero denotes that both
forecasts perform equally.

The *competitive skill score* plot (Fig. 11) includes the comparison between the trained model, the persistence and the ordi-
225 nary least squared regression. The climatological skill scores are calculated separately for each forecast step (lead time) and
summarized as a box-and-whiskers plot over all stations and forecasts with all contributing terms (Fig. 12), and as simplified
version showing the skill score only.

In addition to the statistical model evaluation, MLAir also allows to assess the importance of individual input variables
through bootstrapping of individual input variables. For this, the time series of each individual input variable is resampled $n$
230 times (with replacement) and then fed to the trained network. Afterwards, the skill scores of the bootstrapped predictions are
calculated using the original forecast as reference. If an input variable is important to achieve a good model forecast, it will
thus show up with a large negative skill score in the *bootstrap skill score* plot (Fig. 13). A more detailed description of this
approach is given in Kleinert et al. (2020, submitted).

## 3 Design considerations and choices

In this section we present the general concepts on which MLAir is based. Apart from the choice of the underlying programming
language and the used packages and frameworks (section 3.1), we explain how the concept of *Run Modules* (section 3.2),
*Model Class* (section 3.3) and *Data Handler* (section 3.4) was conceived and how these modules interact with each other.
More detailed information on, for example, how to adapt these modules can be found in the corresponding subsection of the
subsequent section 4.

### 3.1 Coding Language

As underlying coding language *python* (Python Software Foundation, 2018, release 3.6.8) was used for two major reasons. First,
*python* is pretty much independent on the operating system and is not required to be compiled before a run. *python* is flexible
to handle different tasks like data loading from web, training of the ML model or plotting. Numerical operations can executed
quite efficiently due to the fact that they are usually performed by highly optimized and compiled mathematical libraries.
Furthermore, because of its popularity in science and economics, *python* has a huge variety of freely available packages to
use. Secondly, *python* is currently the language in the ML community (Elliott, 2019) and has well-developed easily-to-use
frameworks like *TensorFlow* (Abadi et al., 2015) or *PyTorch* (Paszke et al., 2019) which are state-of-the-art tools to work on
ML problems. Due to the presence of such compiled frameworks, there is for instance no performance loss during the training,
which is the biggest part of the ML workflow, by using python.





Concerning the ML framework, *Keras* (Chollet et al., 2015, release 2.2.4) was chosen for the ML parts using *TensorFlow* (release 1.13.1) as back-end. *Keras* is a framework that abstracts functionality out of its back-end by providing a more simple syntax and implementation. For advanced model architectures and features it is still possible to implement parts or even the entire model in native *TensorFlow* by using the *Keras* front-end for training. Furthermore, *TensorFlow* has GPU support for training acceleration if a GPU device is available on the running system.

For data handling, we chose a combination of *xarray* (Hoyer and Hamman, 2017; Hoyer et al., 2020, release 0.15.0) and *pandas* (Wes McKinney, 2010; Reback et al., 2020, release 1.0.1). *pandas* is an open source tool to analyse and manipulate data primarily designed for tabular data. *xarray* that was inspired by *pandas* is developed to work with multi-dimensional arrays as simple and efficient as possible. *xarray* is based on the off-the-shelf *python* package for scientific computing *NumPy* (van der Walt et al., 2011, release 1.18.1) and introduces labels in form of dimensions, coordinates, and attributes on top of raw
*NumPy*-like arrays.

## 3.2   Run Modules

MLAir models the ML workflow as a sequence of self-contained stages called *Run Modules* that handle distinct tasks whose calculations or results are usually required for all subsequent stages. At run time, all *Run Modules* can interchange information through a temporary data store. All *Run Modules* are executed sequentially upon successful termination of the precursor.
Advanced work flow concepts such as conditional execution of *Run Modules*, are not implemented in this version of MLAir. Also, *Run Modules* cannot be run in parallel, although a single *Run Module* can very well execute parallel code. In the default setup (c.f. Fig. 1), the MLAir workflow constitutes on the following *Run Modules*:

- **Run Environment:** This *Run Module* is the base class for all other *Run Modules*. By wrapping the *Run Environment* around all *Run Modules*, parameters are tracked, the workflow logging is centralized, and the temporary data store is
initialized. After each *Run Module* and at the end of the experiment, *Run Environment* guarantees a smooth (experiment) closure by providing supplementary information on stage execution and parameter access from the data store.

- **Experiment Setup:** Initial stage of MLAir to set up the experiment workflow. Parameters which are not customized are filled with default settings and stored for the experiment workflow. Furthermore, all local paths for the experiment itself but also for data are created during experiment setup.

- **Preprocessing:** During preprocessing, the MLAir loads all required data and carries out typical ML preparation steps to have the data ready-to use for training. If the *Default Data Handler* is used, this step includes downloading or loading of (locally stored) data, data transformation and interpolation. Finally, data are split into the subsets *train*, *val*, and *test*.

- **Model Setup:** The model setup *Run Module* builds the raw ML model implemented as *Model Class* (see section 3.3), sets *Keras* and *TensorFlow* callbacks and checkpoints for the training, and finally compiles the model. Additionally if
using a pre-trained model, the weights of this model are loaded during this stage.





- **Training:** According to the batch size, training and validation data are distributed to properly feed the ML model. Right after, the actual training starts. After each epoch of training, the model performance is evaluated on validation data. If performance improved compared to previous cycles, the model is stored as *best model*. In this way, the final model is the best training model according to validation performance.

- **Postprocessing:** In the final stage, the trained model is statistically evaluated on the test data set. For comparison, MLAir provides two additional forecasts, first an ordinary multi-linear least squared fit trained on the same data like the ML model and second a persistence forecast, where observations of the past represent the forecast for the next steps within the prediction horizon. For daily data, the persistence forecasts refers to the last observation of each sample to hold for all forecast steps. Skill scores based on the model training and evaluation metric are calculated for all forecasts and 290 compared with climatological statistics. The evaluation results are saved as publication-ready graphics. Furthermore, a bootstrapping technique is used to evaluate the importance of each input feature. More details on the statistical analysis that is carried out can be found in section 2.4. Finally, an unpretentious geographical overview map containing all stations is created for convenience.

Ideally this predefined default workflow should meet the requirements for an entire end-to-end ML workflow on station-wise 295 observational data. Nevertheless, MLAir provides options to customize the workflow according to the application needs (see section 4.8).

## 3.3 Model Class

In order to ensure a proper functioning of ML models, MLAir uses a *Model Class*, so that all models are created according to the same scheme. Inheriting from the *Abstract Model Class* guarantees a correct handling during the workflow. The *Model* 300 *Class* is designed to follow an easy plug-and-play behaviour so that within this security mechanism, it is possible to create highly customized models with the frameworks *Keras* and *TensorFlow*. We know that wrapping such a class around each ML model is slightly more complicated, but by requiring the user to build their models in the style of a *Model Class*, the model structure can be documented more easily and there is less potential for errors when interacting with MLAir. More details on the *Model Class* can be found in section 4.5.

## 3.4 Data Handler

In analogy to the *Model Class*, the *Data Handler* organizes all operations related to data retrieval, preparation and provision of a single data origin. For example, if a set of observation stations is being examined in the MLAir workflow, a new instance of the *Data Handler* is created for each station automatically and MLAir will take care of the iteration across all stations. To ensure a smooth integration into MLAir, each *Data Handler* must also follow certain rules. As with the creation of a model, it is 310 not necessary to modify MLAir's source code. Instead, the *Abstract Data Handler* class provides guidance on which methods a *Data Handler* needs to interact smoothly with the workflow.





By default MLAir uses the *Default Data Handler*. It accesses data from JOIN as demonstrated in section 2.2. A detailed description of how to use this *Data Handler* can be found in section 4.4. However, if a different data source or structure is used for an experiment, the *Default Data Handler* must be replaced by a custom *Data Handler* based on the *Abstract Data*

*Handler*. Simply put, such a custom handler requires methods for creating itself at runtime and methods that return the inputs and outputs. Partitioning according to the batch size or suchlike is then handled by MLAir at the appropriate moment and does not need to be integrated into the custom *Data Handler*. Further information about custom *Data Handlers* follows in section 4.3.

## 4 Configuration of Experiment, Data Handler, and Model in the MLAir workflow

Beside the already described workflow adjustments, MLAir offers a high number of configuration options. Instead of defining parameters at different locations inside the code, all parameters are centralized set in the *Experiment Setup*. In this section, we describe all parameters that can be modified and the authors' choices for default settings when using the default workflow of MLAir.

### 4.1 Host System and Processing Units

The MLAir workflow can be adjusted to the hosting system. For that, the local paths for experiment and data are adjustable (see Table 1 for all options). Both paths are separated by choice. This has the advantage that the same data can be used multiple times for different experiment setups if stored outside the experiment path. Contrary to the data path placement, all created plots and forecasts are saved in the experiment path by default, but this can be adjusted through the *plot_path* and *forecast_path* parameter.

Concerning the processing units, MLAir supports both central processing units (CPU) and GPUs. Due to their bandwidth optimization and efficiency on matrix operations, GPUs have become popular for ML applications (c.f., Krizhevsky et al., 2012). Currently, the sample models implemented in MLAir are based on *TensorFlow* v1.13.1, which has distinct branches: the *tensorflow-1.13.1* package for CPU computation and the *tensorflow-gpu-1.13.1* package for GPU devices respectively. Depending on the operating system, the user needs to install the appropriate library if using *TensorFlow* releases 1.15 and older

(TensorFlow, 2020). Apart from this installation issue, MLAir is able to detect and handle both *TensorFlow* versions during run time. An MLAir version to support *TensorFlow* v2 is planned for the future (see section 5).

### 4.2 Preprocessing

In the course of preprocessing, the data are prepared to allow immediate use in training and evaluation without further preparation. In addition to the general data acquisition and formatting, which will be discussed in section 4.3 and 4.4, preprocessing

also handles the splitting into training, validation, and test data. All parameters discussed in this section are listed in Table 2.

Data are split into subsets along the temporal axis and station between a hold-out data set (called test data) and the data that are used for training (resp. training data) and model tuning (validation data). For each subset, a *{train,val,test}_start* and





*{train,val,test}_end* date not exceeding the overall time span (see section 4.4) can be set. Additionally, for each subset it is possible to define a minimal number of available samples per station *{train,val,test}_min_length* to remove very short time series that potentially cause misleading results especially in the validation and test phase. A spatial split of the data is achieved by assigning each station to one of the three subsets of data. The parameter *fraction_of_training* determines the ratio between hold-out data and data for training and validation, where the latter two are always split with a ratio of $80\%$ to $20\%$ which is a typical choice for these subsets.

To achieve absolute statistical data subset independence, data should ideally be split along both temporal and spatial dimension. Since the spatial dependency of two distinct stations may variegate related to weather regimes or season and time of day (Wilks, 2011), a spatial and temporal division of the data might be useful, as otherwise a trained model can presumably lead to over-confident results. On the other hand, by applying a spatial split in combination with a temporal division, the amount of utilizable data can drop massively. In MLAir, it is therefore up to the user to split data either in the temporal or along both dimensions by using the *use_all_stations_on_all_data_sets* parameter.

## 4.3 Custom Data Handler

The integration of a custom *Data Handler* into the MLAir workflow is done by inheritance from the *Abstract Data Handler* class and implementation of at least the *__init__()* method, and the accessors *get_X()*, and *get_Y()*. The custom *Data Handler* is added to the MLAir Workflow as a parameter without initialization. At runtime, MLAir then queries all the required parameters of this custom *Data Handler* from it's arguments and keyword arguments, loads them from the data store and finally calls the constructor. If data need to be downloaded or preprocessed, this should be executed inside the constructor. It is sufficient to load the data in the accessor methods if the data can be used without conversion. We would like to remind that a *Data Handler* is only responsible for a single data origin and the iteration and distribution on batches is taken care of by MLAir.

The accessor methods for input and target data form a clearly defined interface between MLAir and the custom *Data Handler*. During training the data are needed as *NumPy* array, for preprocessing and evaluation the data are partly used as *xarray*. Therefore the accessor methods have the parameter *as_numpy* and should be able to return both formats. Furthermore it is possible to use an individual upsampling technique for training. To activate this feature the parameter *upsamling* can be enabled. If such a technique is not used and therefore not implemented, the parameter has no further effect.

Two other methods do not return a value in the default implementation, but do not necessarily have to be adapted. With the method *transformation()* it is possible to either define or calculate the transformation properties of the *Data Handler* before initialization. The returned properties are then applied to all subdata sets, namely training, validation and testing. Another supporting class method is *get_coordinates()*. This method is currently used only for the map plot for geographical overview (see section 2.3). To feed the overview map, this method must return a dictionary with the geographical coordinates indicated by the keys *lat* and *lon*.





## 4.4 Default Data Handler

In this section we describe a concrete implementation of a *Data Handler*, namely the *Default Data Handler* using data from the JOIN interface in detail.

Regarding the data handling and preprocessing, several parameters can be set to control the choice of inputs, size of data, etc. in the *Data Handler* (see Table 3). First, the underlying raw data is required to load from the web. The current version of the *Default Data Handler* is configured for use with the REST API of the JOIN interface (Schultz et al., 2017c). Alternatively, data

could be already available on the local machine in the directory *data_path*, e.g. from a previous experiment run. Additionally, a user can force MLAir to load fresh data from web by enabling the *overwrite_local_data* parameter. According to the design structure of a *Data Handler*, data are handled separately for each observational station indicated by its ID. By default, the *Default Data Handler* uses all German air quality stations provided by the German *Umweltbundesamt* (UBA) that are indicated as "background" stations according to the *European Environmental Agency* (EEA) Airbase classification (European Parliament

and Council of the European Union, 2008). Using the *stations* parameter, a user-defined data collection can be created. To filter the stations, the parameters *network* and *station_type* can be used as described in Schultz et al. (2017a) and the documentation of JOIN (Schultz et al., 2017c).

For the *Default Data Handler*, it is recommended to specify at least

- the number of preceding time steps to use for a single input sample (*window_history_size*),

- if and which interpolation should be used (*interpolation_method*),

- if and how many missing values are allowed to fill by interpolation (*limit_nan_fill*),

- and how many time steps the forecast model should predict (*window_lead_time*).

Regarding the data content itself, each requested variable must be added to the *variables* list and be part of the *statistics_per_var* dictionary together with a proper statistic abbreviation (see documentation of Schultz et al., 2017c). If not pro-

395 vided, both parameters are chosen from a standard set of variables and statistics. Regarding the target variable, similar actions are required. Firstly, target variables are defined in *target_var*, and secondly, the target variable need also to be part of the *statistics_per_var* parameter. Note that the JOIN REST API calculates these statistics online from hourly values, thereby taking into account a minimum data coverage criterion. Finally, the overall time span the data shall cover can be defined via *start* and *end*, and the temporal resolution of the data is set with *sampling*. At this point, we want to refer to section 5, where we

discuss the temporal resolution currently available.

## 4.5 Defining a Model Class

The idea of using *Model Classes* was already motivated in section 3.3. Here, we show more details on the implementation and customization.

To achieve the goal of an easy plug-and-play behaviour, each ML model implemented in MLAir must inherit from the

405 *Abstract Model Class* and the methods *set_model* and *set_compile_options* are required to be overwritten for the custom model.





Inside *set_model*, the entire model from inputs to outputs is created. Thereby it has to be ensured that the model is compatible with *Keras* to be compiled. MLAir supports both the functional and sequential *Keras* application programming interface. For details on how to create a model with *Keras*, we refer to the official *Keras* documentation (Chollet et al., 2015). All options for the model compilation should be set in the *set_compile_options* method. This method should at least include information

on the training algorithm (*optimizer*), and the loss to measure performance during training and optimize the model for (*loss*). Users can add other compile options like the learning rate (*learning_rate*), *metrics* to report additional merely informative performance metrics, or options regarding the weighting as *loss_weights*, *sample_weight_mode* or *weighted_metrics*. Finally, methods that are not part of *Keras* or *TensorFlow* like customized loss functions or self-made model extensions are required to be added as so-called *custom_objects* to the model so that *Keras* can properly use these custom objects. For that, it is necessary

to call the *set_custom_objects* method with all custom objects as key value pairs. See also the official *Keras* documentation for further general information on custom objects.

An exemplary implementation of a little model using a single convolution and three fully connected layers is shown in Fig. 14. By inheriting from the *Abstract Model Class* (l. 7), invoking of its constructor (l. 13), defining the *set model* (l. 26 - 40) and *set compile options* (l. 42 - 45) method, whereas the call of these both methods (l. 22 - 23), the custom model is immediately

usable for MLAir. Additionally, the loss is added to the custom objects (l. 24). The last step would not be necessary in this case, because an error function incorporated in *Keras* is used (l. 2 / 44). For demonstration purposes of how to use a customized loss, it is added nevertheless.

For another example we refer to Kleinert et al. (2020) who used extensions to the standard *Keras* library in their workflow. So-called inception blocks (c.f. Szegedy et al., 2015) and a modification of the two-dimensional padding layers were implemented

as *Keras* layers and could be used in the model afterwards. In such a case it is important to add the corresponding classes to the *custom_objects*, as mentioned above.

## 4.6 Training

With the parameters *train_model* and *create_new_model* either a halted or interrupted training can be resumed (or extended) or skipped if no training is scheduled since the model was already trained before. Most parameters to set for the training

stage are related to hyperparameter tuning (c.f. Table 4). Firstly, the *batch_size* can be set. Furthermore, the number of *epochs* to train is required to be adjusted. Last but not least, the used *model* itself must be provided to MLAir including additional hyperparameters like the *learning_rate* the algorithm to train the model (*optimizer*) and the *loss* function to measure model performance. For more details on how to implement a ML model properly we refer to section 4.5.

Due to its application focus on meteorological time series and therefore on solving a regression problem, MLAir offers a

particular handling of training data. A popular technique in ML, especially in the image recognition field that usually perform classification tasks, is to assume an independent and identically distribution and therefore augment and randomly shuffle data to produce a larger number of input samples with a broader variety. For meteorological applications, these techniques should be carefully selected, because of the lack of statistical independence of most data and auto correlation (see also Schultz et al., 2020, forthcoming). To avoid generating over-confident forecasts, train and test data are split into blocks so that little or no





overlap remains between the datasets. Another common problem in ML, not only in the meteorological context, is the natural under-representation of extreme values, i.e. an imbalance problem. To address this issue, MLAir allows to place more emphasis on such data points. The weighting of data samples is conducted by an over-representation of values that can be considered as extreme regarding the deviation from a mean state in the output space. This can be applied during training by using the *extreme_values* parameter. For positively skewed distributions, it could be helpful to apply this over-representation only on the

right tail of the distribution (*extremes_on_right_tail_only*). Furthermore, it is possible to shuffle data within, and only within, the training subset randomly by enabling *permute_data*.

## 4.7 Validation

The configuration of the ML model validation is related to the postprocessing stage. As mentioned in section 3.2, in the default configuration there are three major validation steps undertaken after each run besides the creation of graphics: First, the trained

model is opposed to the two reference models, a simple linear regression and a persistence prediction. Second, these models are compared with climatological statistics. Lastly, the influence of each input variable is estimated by a bootstrap procedure.

Due to its encroachment on time or the irrelevance for the custom workflow, the calculation of the input variable sensitivity can be skipped and the graphics creation part can be shortened. To perform the sensitivity study, the parameter *evaluate_bootstraps* must be enabled and the *number_of_bootstraps* defines, how many samples shall be drawn for the evaluation

(c.f. Table 5). If such a sensitivity study was already performed and the training stage was skipped, the *create_new_bootstraps* parameter should be set to *False* to reuse already preprocessed samples if possible. Regarding the creation of graphics, the parameter *plot_list* can be adjusted. If not specified, a default selection of graphics is generated. When using *plot_list*, each graphic to be drawn must be specified individually. More details about all possible graphics have already been provided in section 2.3 and 2.4. In the current version, the validation as part of MLAir's default postprocessing stage cannot be easily

extended, but it is still possible to append another *Run Module* to the workflow to perform an individual validation additionally.

## 4.8 Custom Run Modules and Workflow Adaptions

MLAir offers the possibility to define and execute a custom workflow for situations in which special calculations or data evaluation not available in the standard version are to be performed. For this purpose it is not necessary to modify the program code of MLAir, but instead user-defined *Run Modules* can be included in a new workflow. This is done analogous to the

465 procedure of *Model Class* by inheritance from the base class *Run Environment* and the individually adapted programming of a *Run Module*. Compared to the very simple examples from section 2, such a use of MLAir requires a slightly increased effort. The implementation of the *Run Module* is done straightforwardly by a constructor method, which initializes the module and executes all desired calculation steps upon call. To execute the custom workflow, the MLAir *Workflow* class must be loaded and then each *Run Module* must be registered. The order in which the individual stages are added determines the execution

sequence.

As custom workflows will generally be necessary if a custom *Run Module* is to be defined, we briefly describe how the central data store mentioned in section 3.2 interacts with the workflow module. With the data store it is possible to share any





kind of information from previous or subsequent stages. By invoking the constructor of the super class during the initialization of a custom *Run Module*, the data store is automatically connected with this module. Information can then be set or queried

using the accesssor methods *get* and *set*. For each saved information object a separate namespace called *scope* can be assigned. If not specified, the object is always stored in the general scope. If the scope is specified, a separate sub-scope is created. Information stored in this scope memory cannot be accessed from the general scope memory, but conversely all sub-scopes have access to the general scope. For example, more general objects can be set in the general scope and objects specific to a sub-data set, such as test data, can be stored in under the scope *test*. If some objects for the keyword *test* are retrieved from the

data store, then for non-existent objects in the *test* namespace attributes from the general scope are used if available.

An example for the implementation of a custom *Run Module* embedded in a custom *Workflow* can be found in Fig. 15. The custom *Run Module* named *CustomStage* inherits from the base class *Run Environment* (l. 4) and calls its constructor (l. 8) on initialization. The *CustomStage* expects a single parameter (*test_string*, l. 7), that is used during the *run* method (l. 11 - 15). The *run* method first logs two information messages by using the *test_string* parameter (l. 12 - 13). Then it extracts the value

of the parameter *epochs* (l. 14) that has been set in the *Experiment Setup* (l. 21) from the data store and logs the value of this parameter too. To run this custom *Run Module* is has to be included in a *Workflow*. First an empty workflow is created (l. 19) and then individual *Run Modules* are attached (l. 21 - 23). As last step, this new defined workflow is executed by calling the *run* method (l. 25).

### 4.9   How to continue an experiment?

There can be different reasons for the continuation of an experiment. First of all, by looking at the monitoring graphs, it could be discovered that training has not yet converged and the number of epochs should be increased. Instead of training a new network from scratch, the training can be resumed from the latest epoch to save time. To do so, the parameter *epochs* must be increased accordingly and *create_new_model* must be set to *False*. If the *model* output folder has not been touched, the intermediate results and the history of the previous training are usually available in full, so that MLAir can continue the training as if it

had never been interrupted. Another reason for a continuation would be the interruption of the training for unexpected reasons such as runtime exceedance on batch systems. By keeping the same number of epochs and switching off the creation of a new model, the training continues at the last checkpoint (see Model Setup in section 3.2).

### 5   Limitations

Even though MLAir addresses a wide range of ML related problems and allows embedding of many different ML architectures

and customized workflows, it is still no universal Swiss Army knife, but focuses on the application of station time series forecasting. In this section we will explain the limitations of MLAir and why MLAir ends at these points.

Due to the scientifically oriented development of MLAir starting from a specific research question (Kleinert et al., 2020), MLAir could initially only use data from the REST API of JOIN. This binding has already been revoked in the current version, however, the *Default Data Handler* still uses this data source. Furthermore, MLAir always expects a particular structure in the



data and especially considers the data as a collection of time series data from various stations. We are currently investigating the possibility of integrating grid data, which could be taken from a weather model, and timeless data such as topography into the MLAir workflow, but cannot yet present any results on how easy such an integration would be.

While MLAir can technically handle data in different time resolutions, it has been tested primarily on daily aggregated data due to the specific science case which served as seed for its development. The use of different temporal resolutions was spot-checked and could be successfully confirmed without obvious errors, but we cannot guarantee that the results will be meaningful if data in other temporal resolutions are used as inputs. In particular, most of the evaluation routines may not make sense for data in less than hourly or greater than daily resolution. Note also that MLAir does not perform explicit error checking or missing value handling. Such functionality must be implemented within the *Data Handler*. MLAir expects a ready-to-use data set without missing values provided by the *Data Handler* during training.

Another limitation is the choice of the underlying libraries and their versions. Due to the selection of *TensorFlow* as backend, it is not possible to use *PyTorch* or other frameworks in combination with MLAir. Specifically, MLAir was developed and tested with *TensorFlow* version 1.13.1, as the HPC systems on which our experiments are performed support this version. We have already tested MLAir occasionally with the *TensorFlow* version 1.15 and could not find any errors. But due to the lack of extensive testing, we can therefore not make any reliable statement about the functionality with newer versions like 1.15 or 2.X yet. It is planned to implement an updated version of MLAir with the new *TensorFlow* version 2.X as soon as our systems support this version without any problems.

## 6 Summary

MLAir is an innovative software package intended to facilitate high-quality meteorological studies using ML. By providing an end-to-end solution based on a specific scientific workflow of time series prediction, MLAir enables a transparent and reproducible conduction of ML experiments in this domain. Due to the plug-and-play behaviour it is straightforward to explore different model architectures and change various aspects of the workflow or model evaluation. Since MLAir is based on a pure python environment, it is highly portable. It has been tested on various computing systems from desktop workstations to high-end supercomputers.

MLAir is under continuous development. Further enhancements of the program are already planned and can be found in the issue tracker (see annex code availability). Ongoing developments concern the extension of the statistical evaluation methods, the graphical presentation of the results and the flawless support of temporal resolutions other than daily aggregated data. Through further code refactoring, MLAir will become even more versatile as the decoupling of individual components is being pushed forward. In particular, it is planned to structure the data handling in a more modular way so that varying structured data sources can be connected and used without much effort. We invite the community of meteorological ML scientists to participate in the further development of MLAir through comments and contributions to code and documentation. A good starting point for contributions is the issue tracker of MLAir.



Even if MLAir cannot be the all-encompassing environment for every kind of meteorological ML problem, we hope that MLAir can serve as a blueprint for application developments in this field, as it seeks to combine best practices from ML with best practices of meteorological model evaluation and data preprocessing. MLAir is thus a contribution to strengthen the
540 integration of the communities of ML and meteorology or air quality research.

*Code availability.* The current version of MLAir is available from the project website https://gitlab.version.fz-juelich.de/toar/mlair under the MIT licence. The exact version v1.0.0 of MLAir described in this paper and used to produce the shown code examples is archived on B2SHARE (http://doi.org/10.34730/fcc6b509d5394dad8cfdfc6e9fff2bec). Detailed installation instructions are provided in the project page readme file. There is also a Jupyter notebook with all code snippets to reproduce the examples highlighted in this paper.

*Author contributions.* **Lukas H. Leufen:** Conceptualization, Investigation, Methodology, Software, Validation, Visualization, Writing – original draft preparation. **Felix Kleinert:** Conceptualization, Investigation, Methodology, Software, Validation, Writing – review & editing. **Martin G. Schultz:** Funding Acquisition, Project Administration, Supervision, Writing – review & editing.

*Competing interests.* The authors declare that they have no conflict of interest.

*Acknowledgements.* **Jülich Supercomputing Centre:** Resources. **European Research Council:** Financial Support (Horizon 2020, *IntelliAQ - Artificial Intelligence for Air Quality, 787576*)



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





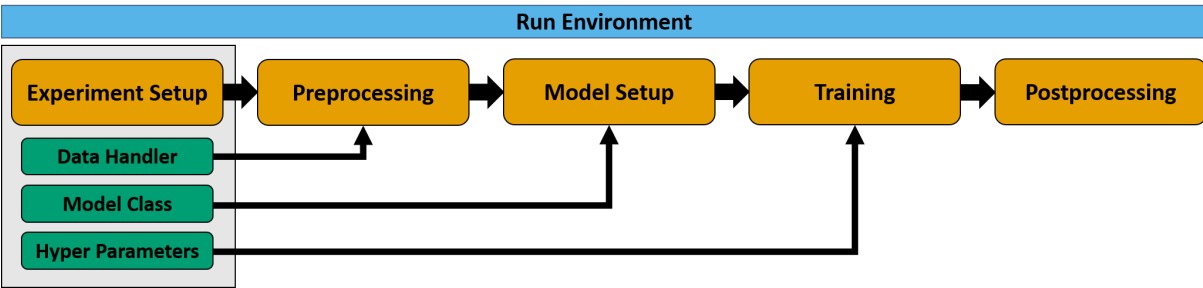

**Figure 1.** Visualization of the MLAir standard setup *DefaultWorkflow* including the stages *Experiment Setup*, *Preprocessing*, *Model Setup*, *Training*, and *Postprocessing* (all highlighted in orange) embedded in the *Run Environment* (sky blue). Each Experiment customization (bluish green) like the exemplary shown *Data Handling*, *Model Class*, and *Hyperparameter* is set during the initial *Experiment Setup* and do affect various stages of the workflow.





```
1  import mlair
2
3  # just give it a dry run without any modification
4  mlair.run()
```

```
INFO: DefaultWorkflow started
INFO: ExperimentSetup started
INFO: Experiment path is: /home/<usr>/mlair/testrun_network
...
INFO: load data for DEBW107 from JOIN
INFO: load data for DEBY081 from JOIN
INFO: load data for DEBW013 from JOIN
INFO: load data for DEBW076 from JOIN
INFO: load data for DEBW087 from JOIN
...
INFO: Training started
...
INFO: DefaultWorkflow finished after 0:03:04 (hh:mm:ss)
```

**Figure 2.** A very simple Python script (e.g. written in a *Jupyter Notebook* or Python file) calling the MLAir package without any modification. Selected parts of the corresponding logging of the running code are shown underneath. Results of this and following code snippets have to be seen as a pure demonstration, because the default neural network is very simple.





```
import mlair
# our new stations to use
stations = ['DEBW030', 'DEBW037', 'DEBW031', 'DEBW015', 'DEBW107']
# expanded temporal context to 14 (days, because of default
         sampling="daily")
window_history_size = 14
# restart the experiment with little customisation
mlair.run(stations=stations,
window_history_size=window_history_size)
```

```
INFO: DefaultWorkflow started
INFO: ExperimentSetup started
...
INFO: load data for DEBW030 from JOIN
INFO: load data for DEBW037 from JOIN
INFO: load data for DEBW031 from JOIN
INFO: load data for DEBW015 from JOIN
...
INFO: Training started
...
INFO: DefaultWorkflow finished after 00:02:03 (hh:mm:ss)
```

**Figure 3.** The MLAir experiment has now minor adjustments for the parameters *stations* and *window_history_size*.





```
 import mlair
 # our new stations to use
 stations = ['DEBY002', 'DEBY079']
 # same setting for window_history_size
 window_history_size = 14
 # run experiment without training
mlair.run(stations=stations,
window_history_size=window_history_size,
create_new_model=False,
train_model=False)
```

```
INFO: DefaultWorkflow started
...
INFO: No training has started, because train_model parameter was false.
...
INFO: DefaultWorkflow finished after 0:01:27 (hh:mm:ss)
```

**Figure 4.** Experiment run without training. For this, it is required to have an already trained model in the experiment path.



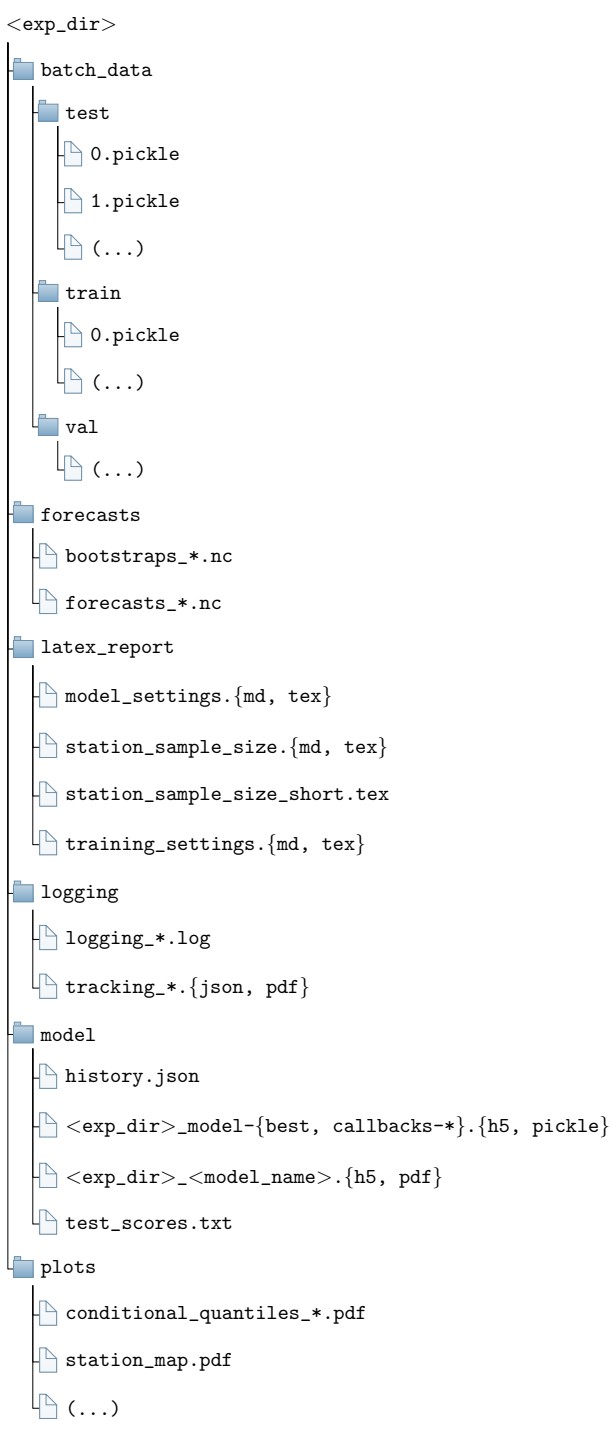

**Figure 5.** Default structure of each MLAir experiment with the subfolders *forecasts*, *latex_report*, *logging*, *model*, and *plots*. *<exp_dir>* is a placeholder for the actual name of the experiment.



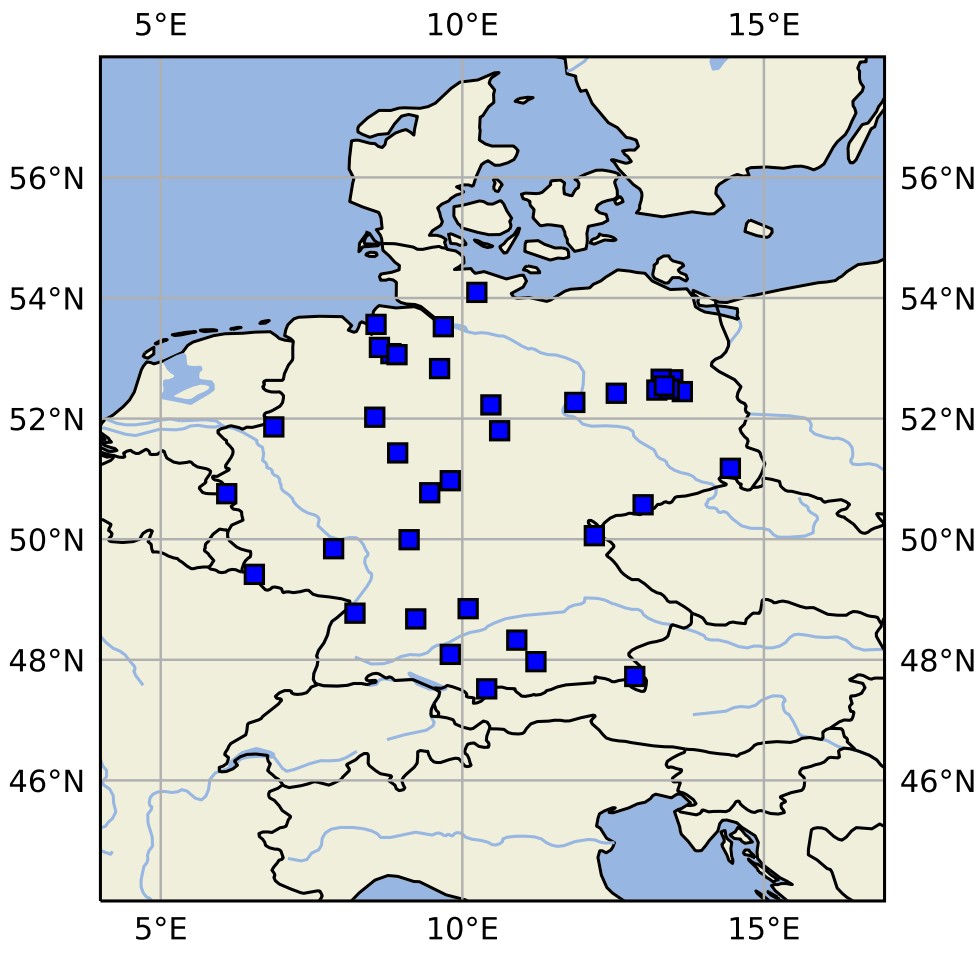

**Figure 6.** Map of central Europe showing the locations of exemplary measurement stations as blue squares.



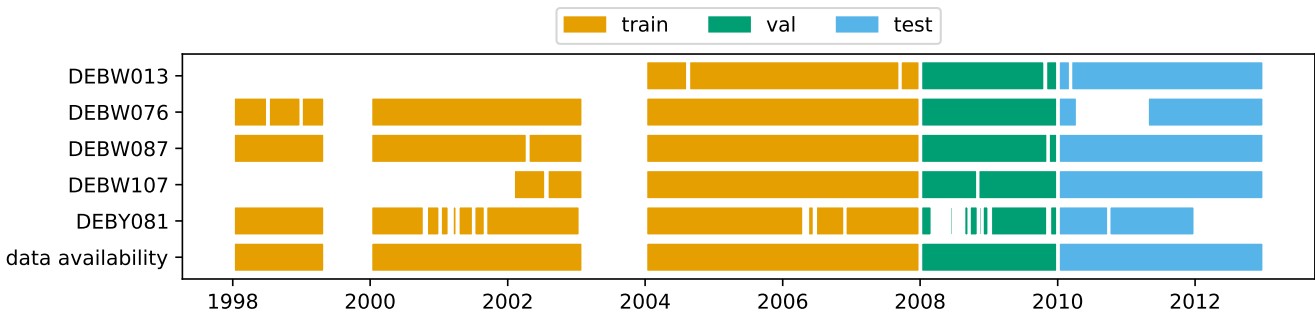

**Figure 7.** Data availability diagram showing the available data for five exemplary measurement stations. The different colours denote which period of the time series is used for the training (orange), validation (green) and test (blue) data set. "data availability" denotes if any of the above mentioned stations has a data record for a given time.





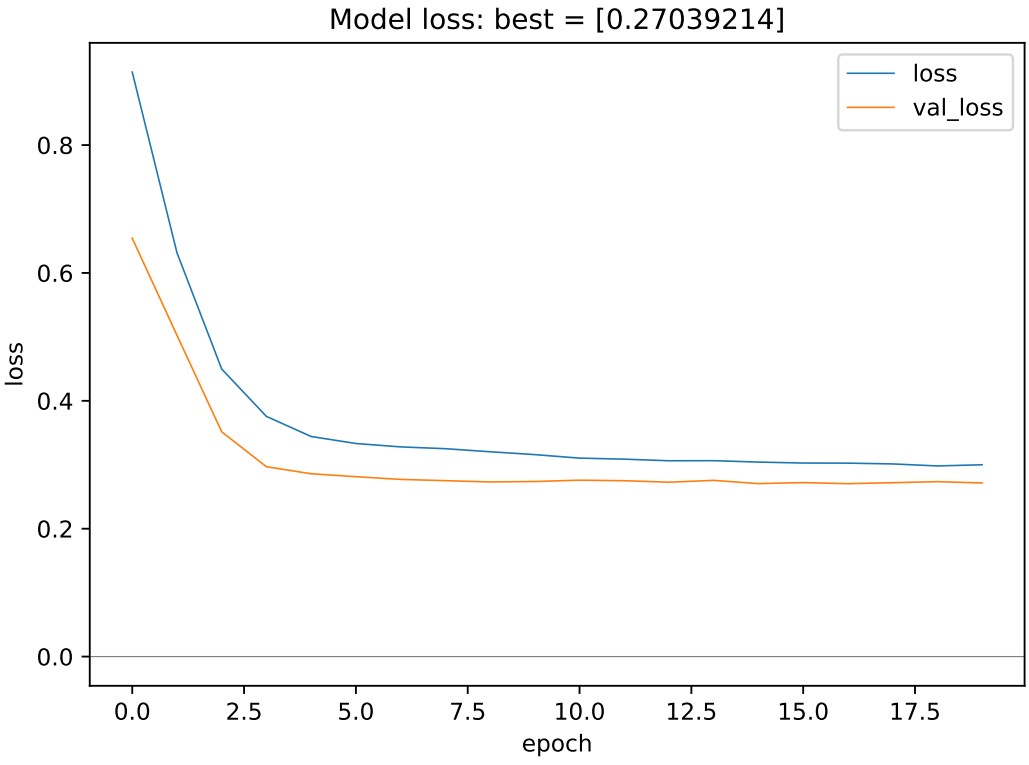

**Figure 8.** Monitoring plots showing the evolution of train and validation loss as a function of the number of epochs. This plot type is kept very simplistic by choice. The underlying data are saved during the experiment so that it would be easy to create a more advanced plot using the same data.



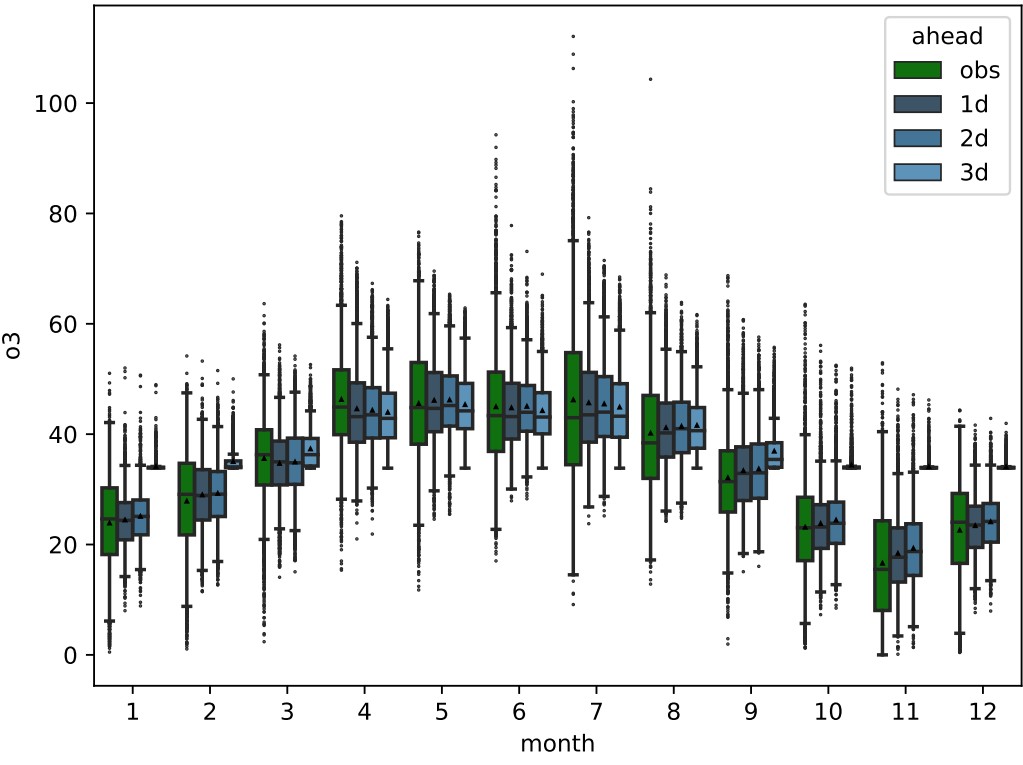

**Figure 9.** Monthly summary plot showing the observations (green) and the predictions for all forecast steps (dark to light blue) separated for each month.



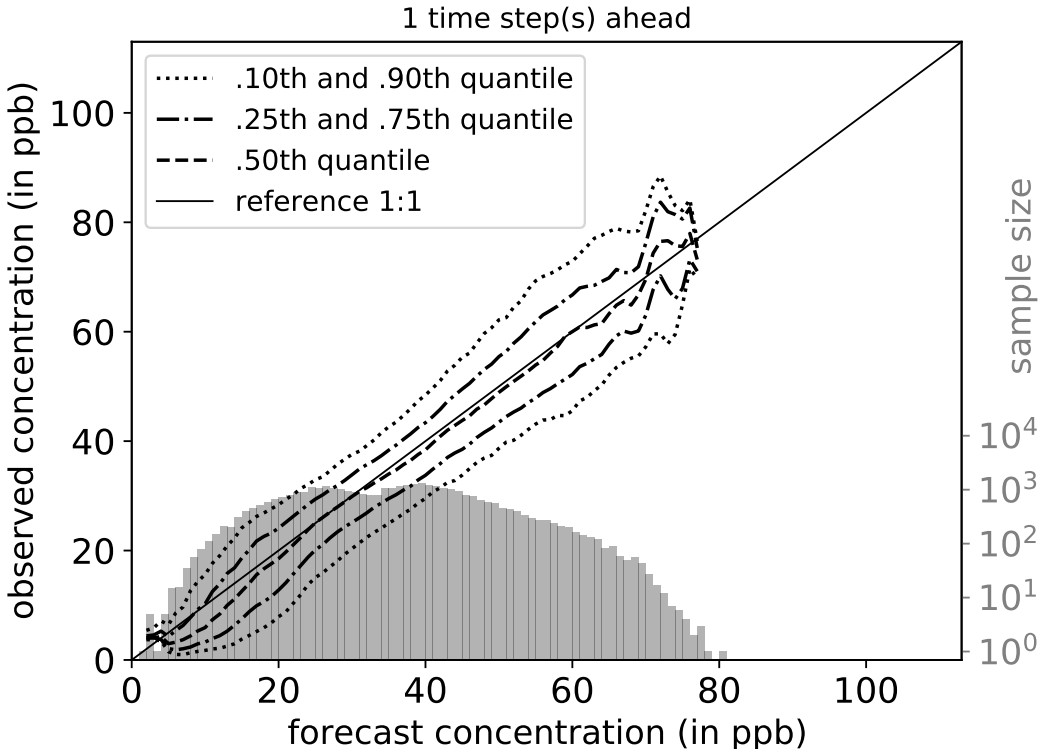

**Figure 10.** Conditional quantiles in terms of calibration-refinement factorization for the first lead time and the full test period. The marginal forecasting distribution is shown as log-histogram in light grey (counting on right axis). The conditional distribution (calibration) is shown as percentiles in different line styles. Calculations are done with a bin size of 1 ppb. Moreover, the percentiles are smoothed by a rolling mean of window size three. This kind of plot was originally proposed by Murphy et al. (1989).



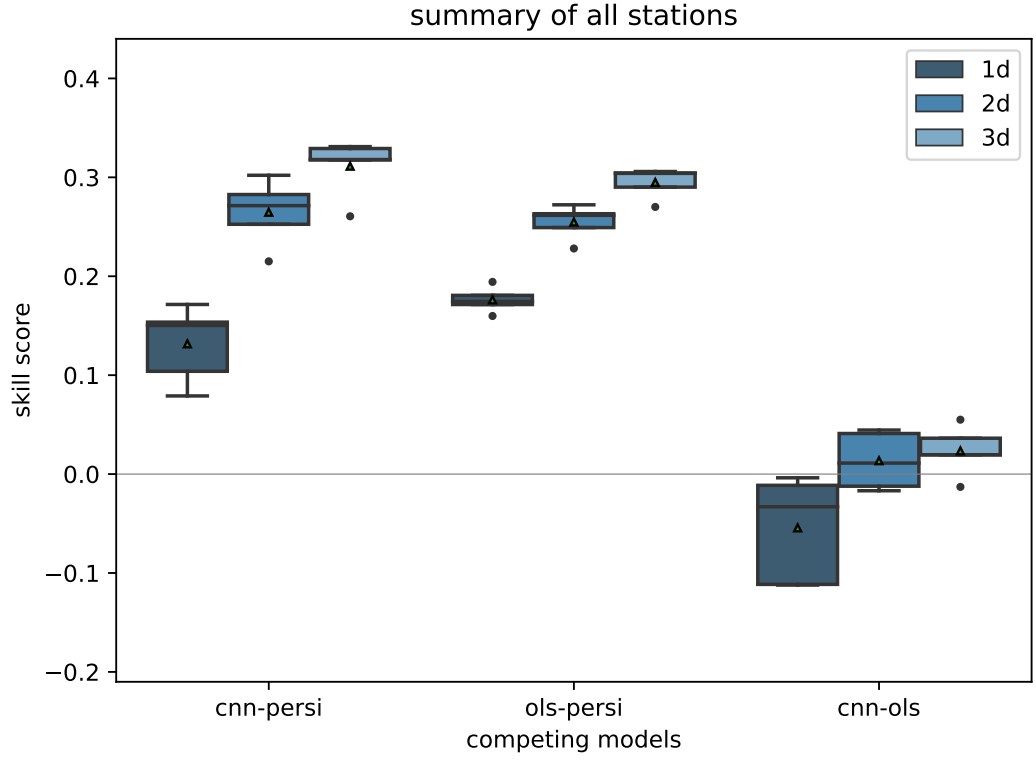

**Figure 11.** Skill scores of different reference models like persistence (persi), and ordinary least square (ols). Skill scores are shown separately for all forecast steps (dark to light blue).





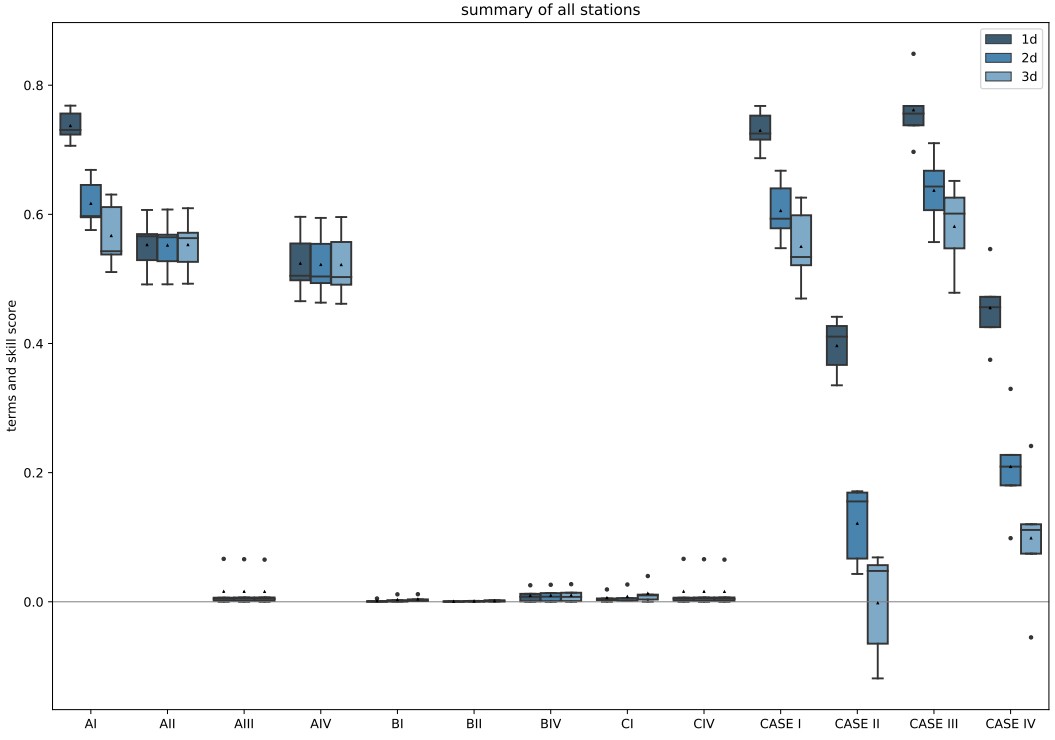

**Figure 12.** Climatological skill scores (CASE I to IV) and related terms of the decomposition as proposed in Murphy (1988). Skill scores and terms are shown separately for all forecast steps (dark to light blue).



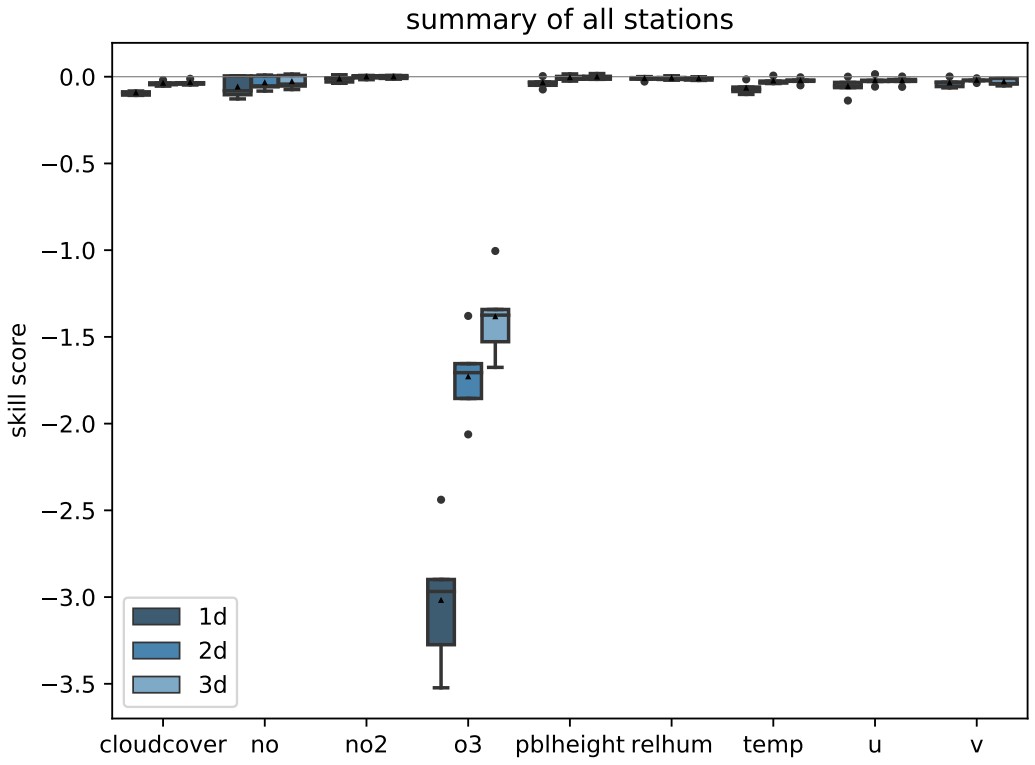

**Figure 13.** Skill score of bootstrapped model input predictions separated for each input variable (x-axis) and forecast steps (dark to light blue) having the original (non-bootstrapped) predictions as reference.





```
import keras
from keras.losses import mean_squared_error as mse
from keras.optimizers import SGD
from mlair.model_modules import AbstractModelClass
from mlair.workflows import DefaultWorkflow
class MyCustomisedModel(AbstractModelClass):
"""
A customised model with a 1x1 Conv, and 2 Dense layers (16,
output shape). Dropout is used after Conv layer.
"""
def __init__(self, input_shape: list, output_shape: list):
# set attributes _input_shape and _output_shape
super().__init__(input_shape[0], output_shape[0])
# apply to model
self.set_model()
self.set_compile_options()
self.set_custom_objects(loss=self.compile_options['loss'])
def set_model(self):
x_input = keras.layers.Input(shape=self._input_shape)
x_in = keras.layers.Conv2D(32, (1, 1))(x_input)
x_in = keras.layers.PReLU()(x_in)
x_in = keras.layers.Flatten()(x_in)
x_in = keras.layers.Dropout(0.1)(x_in)
x_in = keras.layers.Dense(16)(x_in)
x_in = keras.layers.PReLU()(x_in)
x_in = keras.layers.Dense(self._output_shape)(x_in)
out = keras.layers.PReLU()(x_in)
self.model = keras.Model(inputs=x_input, outputs=[out])
def set_compile_options(self):
self.initial_lr = 1e-2
self.optimizer = SGD(lr=self.initial_lr, momentum=0.9)
self.loss = mse
self.compile_options = {"metrics": ["mse", "mae"]}
# Make use of MyCustomisedModel within the DefaultWorkflow
workflow = DefaultWorkflow(model=MyCustomisedModel, epochs=2)
workflow.run()
```

**Figure 14.** Example how to create a custom ML model implemented as *Model Class*. *MyCustomisedModel* has a single 1x1 convolution layer followed by two fully connected layers with a neuron size of 16, and the number of forecast steps. The model itself is defined in the *set_model* method whereas compile options as the optimizer, loss and error metrics are defined in *set_compile_options*. Additionally for demonstration, the loss is added as custom object which is not required because a *Keras* built-in function is used as loss.





```
import mlair
import logging
class CustomStage(mlair.RunEnvironment):
"""A custom MLAir stage for demonstration."""
def __init__(self, test_string):
super().__init__()  # always call super init method
self._run(test_string)  # call a class method
def _run(self, test_string):
logging.info("Just running a custom stage.")
logging.info("test_string = " + test_string)
epochs = self.data_store.get("epochs")
logging.info("epochs = " + str(epochs))
# create your custom MLAir workflow
CustomWorkflow = mlair.Workflow()
# provide stages without initialisation
CustomWorkflow.add(mlair.ExperimentSetup, epochs=128)
# add also keyword arguments for a specific stage
CustomWorkflow.add(CustomStage, test_string="Hello World")
# finally execute custom workflow in order of adding
CustomWorkflow.run()
```

```
INFO: Workflow started
...
INFO: ExperimentSetup finished after 00:00:12 (hh:mm:ss)
INFO: CustomStage started
INFO: Just running a custom stage.
INFO: test_string = Hello World
INFO: epochs = 128
INFO: CustomStage finished after 00:00:01 (hh:mm:ss)
INFO: Workflow finished after 00:00:13 (hh:mm:ss)
```

**Figure 15.** Embedding of a custom *Run Module* in a modified MLAir workflow. In comparison to figures 2, 3, and 4, this code example works on a single step deeper regarding the level of abstraction. Instead of calling the run method of MLAir, the user needs to add all stages individually and is responsible for all dependencies between the stages. By using the *Workflow* class as context manager, all stages are automatically connected with the result that all stages can easily be plugged in.





**Table 1.** Summary of all parameters related to the host system that are required, recommended, or optional to adjust for a custom experiment workflow.

| Host System | | |
| --- | --- | --- |
| Parameter | Default | Adjustment |
| *experiment_date* | testrun | recommended |
| *experiment_name* | {*experiment_date*}_network | — * |
| *experiment_path* | ⟨cwd**⟩/{*experiment_name*} | optional |
| *data_path* | ⟨cwd**⟩/*data* | optional |
| *bootstrap_path* | ⟨data_path⟩/*bootstraps* | optional |
| *forecast_path* | ⟨experiment_path⟩/*forecasts* | optional |
| *plot_path* | ⟨experiment_path⟩/*plots* | optional |

\* only adjustable via the *experiment_date* parameter

\*\* refers to the linux command to get the path name of the current working directory.





**Table 2.** Summary of all parameters related to the preprocessing that are required, recommended, or optional to adjust for a custom experiment workflow.

| Preprocessing | | |
|---|---|---|
| Parameter | Default | Adjustment |
| *stations* | default stations* | recommended |
| *data_handler* | DefaultDataHandler | optional |
| *fraction_of_training* | 0.8 | optional** |
| *use_all_stations_on_all_data_sets* | True | optional |

* default stations: DEBW107, DEBY081, DEBW013, DEBW076, DEBW087

** not used in the default setup because *use_all_stations_on_all_data_sets* is True





**Table 3.** Summary of all parameters related to the default data handler that are required, recommended, or optional to adjust for a custom experiment workflow.

| Default Data Handler | | |
|---|---|---|
| Parameter | Default | Adjustment |
| *data_path* | see Table 1 | optional |
| *stations* | default stations* | recommended |
| *network* | - | optional |
| *station_type* | - | optional |
| *variables* | default variables** | recommended |
| *statistics_per_var* | default statistics** | recommended |
| *target_var* | o3 | recommended |
| *start* | 1997-01-01 | recommended |
| *end* | 2017-12-31 | recommended |
| *sampling* | daily | optional |
| *window_history_size* | 13 | recommended |
| *interpolation_method* | linear | optional |
| *limit_nan_fill* | 1 | optional |
| *min_length**** | 0 | optional |
| *window_lead_time* | 3 | recommended |
| *overwrite_local_data* | False | optional |

\* default stations: DEBW107, DEBY081, DEBW013, DEBW076, DEBW087

\*\* default variables (statistics): o3 (dma8eu), relhum (average_values), temp
(maximum), u (average_values), v (average_values), no (dma8eu), no2 (dma8eu),
cloudcover (average_values), pblheight (maximum)

\*\*\* indicates the required minimum number of samples per station





**Table 4.** Summary of all parameters related to the training that are required, recommended, or optional to adjust for a custom experiment workflow.

| Training | | |
|---|---|---|
| Parameter | Default | Adjustment |
| *train_model* | False | recommended* |
| *create_new_model* | False | recommended* |
| *batch_size* | 512 | optional |
| *epochs* | - | required |
| *loss*** | - | required |
| *metrics*** | - | optional |
| *model* | vanilla model*** | required |
| *learning_rate*** | - | required |
| *optimizer*** | - | required |
| *extreme_values* | - | optional |
| *extremes_on_right_tail_only* | False | optional |
| *permute_data* | False | optional |

\* Note: Both parameters are disabled per default to prevent unintended overwriting of a model. If, upon reversion, these parameters aren't enabled on first execution of a new experiment without providing a suitable and trained ML model, the MLAir workflow is going to fail.

\*\* These parameters are set in the *Model Class*.

\*\*\* As default, a vanilla feed-forward neural network architecture will be loaded for workflow testing. The usage of such a simple network for a real application is at least questionable.





**Table 5.** Summary of all parameters related to the evaluation that are required, recommended, or optional to adjust for a custom experiment workflow.

| Evaluation | | |
|---|---|---|
| Parameter | Default | Adjustment |
| *plot_list* | default plots* | optional |
| *evaluate_bootstraps* | True | optional |
| *number_of_bootstraps* | 20 | optional |
| *create_new_bootstraps* | False** | optional |

\* default plots are: *PlotMonthlySummary*, *PlotStationMap*,
*PlotClimatologicalSkillScore*, *PlotTimeSeries*, *PlotCompetitiveSkillScore*,
*PlotBootstrapSkillScore*, *PlotConditionalQuantiles*, and *PlotAvailability*.
\*\* is automatically enabled if parameter *train_model* (see Table 4) is
enabled.