# Peer review of "MLAir (v1.0) - a tool to enable fast and flexible machine learning on air data time series"

_Geoscientific Model Development, 2020_

## Short Comment (SC1) · 26 Oct 2020

"For example, its data preparation step acknowledges the auto-correlation which is typically seen in meteorological time series"

if a time-series is not random white noise, it is auto-correlated to some extent. Never understood why this is often referred to as being detrimental – e.g. a sine wave is auto-correlated and this is good, because it allows it to be predicted.

---

## Referee Comment (RC1) · Anonymous Referee #1 · 23 Nov 2020

**1   Overview**

The manuscript describes a library which facilitates the development of end-to-end neural network workflows (though the name suggests it may support other ML algorithms) for time series forecasting (mostly focused on air quality).

Though the use case is fairly narrow in scope, the architecture of MLAir and the various features to validate the models using standard meteorological metrics is very interesting. MLAir's design also allows for development of reproducible ML pipelines which is essential if ML techniques are to be more widely used by the climate science community.

[Figure]

Overall, the aims and implementation of MLAir serve as a good template for future development of such frameworks and I consider it a valuable contribution to the geoscientific community.

That being said, it seems to me that certain parts of the manuscript could be improved to enhance readability.

**2  Major Comments**

1. It would be helpful if the abstract clarified that MLAir is focused on neural networks and not any other kind of ML algorithm.

2. It would really help if the manuscript used different styles for conceptually different things. For instance, libraries (e.g., *TensorFlow*) are italicized, and so are class names (e.g., *Data Handler*). It would be preferable if the typewriter font is used to strengthen the correspondence with the figures (`DataHandler`). If *TensorFlow* is italicized, why is MLAir not italicized?

   If names like *Data Handler* refer to class names, then it is unclear why the names are split. If it describes the function performed by a piece of code, then it is not clear why it is italicized.

   Similar issues are found in the caption of Figure 1. Are *Experiment Setup, Preprocessing, Model Setup* class names or verbs? Is *Hyperparameter* a class name or terminology from ML? If it is the latter, why is it italicized, and why if it is the former, then why is it not mentioned in Line 91? Similarly, the text used in the various bubbles in Figure 1 itself could benefit from such formatting (is *Run Environment* a class name or a simply a description?).

   As it stands, the manuscript's treatment of different terminologies is too confusing for the reader to precisely understand what each term stands for. The manuscript
would also benefit from a sentence or two stating the typographical choices made by the authors.

3. Section 2.1 would benefit from a definition of the main components – *Run Environment, Data Handler, Model Class* – what they do, and how they fit into the design of MLAir. Currently, only *Run Environment* is described, and *Data Handler* has a fairly confusing description (Line 93: Data Handler is responsible for an accurate use of the data).

   I strongly recommend rewriting Section 2.1 with the reader in mind, to ensure that readers are systematically introduced to the concepts and/or stages underlying MLAir and the corresponding software components that implement these concepts and/or stages.

4. To ensure the a reader with no background in ML is able to relate to the design of MLAir, a brief introduction to a typical ML workflow (test, train, validation, epoch, hyperparameter, etc.,) would be really helpful in Section 2.1.

5. In Section 2.2, *window_history_size* is not defined anywhere. Therefore, what it does and how it might impact network architecture (Line 128–129) is not clear.

6. Line 142: The term epoch is used without defining it anywhere. Definitions and motivations for such terms could be added to the brief description of a typical ML workflow that was mentioned previously.

7. Line 220: If the skill score is defined as a ratio of a metric like MSE, how does one obtain positive and negative skill?

8. Line 229: The process of bootstrapped predictions could be explained a little more in detail for completeness. "the time series of each individual input variable is resampled n times (with replacement) and then fed to the trained network." was not sufficient for me (at least) to understand what was being attempted and achieved.

9. Figure 12: I might have missed something, but it is unclear what AI, BI, CI, CASEI etc., are. Furthermore, it is unclear what "terms" and "contributing terms" (Line 226) are.

10. Lines 417-421: The line numbers in Figure 14 that are referred to here don't seem to correspond to their respective descriptions. I hope the authors verify the same for other such figures as well.

11. Line 440: A more detailed description of how the imbalance problem is addressed would benefit the reader who wishes to use this feature.

**3 Minor Comments**

1. Line 58: It also allows to deploy → It also allows deploying

2. Line 59: use of GPUs . I think it is proper to acknowledge that usage of GPUs is due to the underlying tensorflow library.

3. Line 60: Concurrent to a simple usage with low barriers for ML-callow scientists,: **Not sure what is meant by this sentence**.

4. Line 84: as many customization → To facilitate customization(?)

5. Line 83: Why is Workflow capitalized?

6. Figure 1: exemplary: **exemplary usually means commendable. I don't think that is what the authors meant.**

7. Figure 6: exemplary measurement stations: **Same as above. This applies to all other places where exemplary has been used.**

[Figure]

8. Line 110: daily aggregated: **does this mean daily mean? daily max? It is unclear how the aggregation is done.**

9. Line 168: Beside the file, which contain the model → Besides the file, which contains the model

10. Line 181: intended to add own graphics in MLAir → intended to add custom graphics in MLAir

11. Line 185: major waters → major water bodies

12. Line 191: are meant to give an insight into → are meant to provide insight into

13. Line 195: each month separately as box-and-whisker → each month separately as a box-and-whisker diagram

14. Line 205: marginal distribution is shown as histogram (lite grey) → marginal distribution is shown as a histogram (light grey)

15. Line 242: independent on the OS → independent of the OS

16. Line 350: Since the spatial dependency of two distinct stations may variegate related → Since the spatial dependency of two distinct stations may vary(?) related

17. Line 433: how to implement a ML model → how to implement an ML model

18. Line 436: is to assume an independent and identically distribution and therefore augment and randomly shuffle data to produce a larger number of input samples with a broader variety. → is to assume independent and identically distributed data and therefore augment and randomly shuffle the data to produce a larger number of input samples with a broader variety. **Consider rewriting this sentence to enhance readability.**

19. Line 442: To address this issue, MLAir allows to place more → To address this issue, MLAir allows placing more

---

## Referee Comment (RC2) · Anonymous Referee #2 · 11 Dec 2020

**1   OVERVIEW**

Through this manuscript, a new library specialized in Neural Networks for air quality forecasting is presented. In general terms, it is well written, the objectives are clear, and main features are correctly established.

Given that Neural Networks are gaining importance and widening their public, it is of high interest developing new tools that facilitate the intersection between this concrete field and any other area. In that sense, this work is an example of transparency and reproducibility, especially important in the Artificial Intelligence domain.

[Figure]

**2 COMMENTS**

1. The title is at some degree confusing. Using the term "Machine Learning" might lead to confusion, as the framework is solely based on deep learning.

2. It feels like section 2 and 3 could be exchanged: before explaining how the framework is used, it might be interesting to know how the framework works. Also, the description of the framework is quite scarce and for a non-computer scientist reader it could be hard to follow.

3. Given that the manuscript aims to improve synergies between air quality experts/practitioners and deep learning methodologies, some kind of introduction to deep learning main points is recommendable.

4. Some terms are not described before being used. Again, as in Section 2 the functioning and experimentation procedure is presented without any previous explanation of the general framework, sometimes it is confusing.

5. Grammar and typo errors should be carefully checked throughout the entire manuscript.

---

## Author Comment (AC1) · 8 Jan 2021

**1   General statement**

We would like to thank the two referees for their review of our manuscript and the many very helpful comments and suggestions for improvement. We would also like to thank Christoph Knote for being the topical editor and for his intensive search for referees. In this document, we will address all the referees' comments one by one and present the modifications made to the manuscript. For better clarity, our responses and changes in the text that refer to Referee #1's comments are highlighted in blue. For responses and amendments related to Referee #2 comments, the highlighting is in red. Other adjustments without a direct relation to one of the referees are highlighted in magenta.

[Figure]

Line numbers given in this document always refer to the peer-reviewed document and not to the corrected version.

**2 Answer to Anonymous Referee #1**

"The manuscript describes a library which facilitates the development of end-to-end neural network workflows (though the name suggests it may support other ML algorithms) for time series forecasting (mostly focused on air quality).

Though the use case is fairly narrow in scope, the architecture of MLAir and the various features to validate the models using standard meteorological metrics is very interesting. MLAir's design also allows for development of reproducible ML pipelines which is essential if ML techniques are to be more widely used by the climate science community.

Overall, the aims and implementation of MLAir serve as a good template for future development of such frameworks and I consider it a valuable contribution to the geoscientific community.

That being said, it seems to me that certain parts of the manuscript could be improved to enhance readability."

Major Comments

1. "It would be helpful if the abstract clarified that MLAir is focused on neural networks and not any other kind of ML algorithm."
   We have added that MLAir is focusing on shallow and deep neural networks.
   "With *MLAir* (Machine Learning on Air data) we created a software environment that simplifies and accelerates the exploration of new machine learning (ML) models, specifically shallow and deep neural networks, for the analysis and forecasting of meteorological and air quality time series." (l.1)

2. "It would really help if the manuscript used different styles for conceptually different things. For instance, libraries (e.g., TensorFlow) are italicized, and so are class names (e.g., Data Handler). It would be preferable if the typewriter font is used to strengthen the correspondence with the figures (DataHandler). If TensorFlow is italicized, why is MLAir not italicized?

If names like Data Handler refer to class names, then it is unclear why the names are split. If it describes the function performed by a piece of code, then it is not clear why it is italicized.

Similar issues are found in the caption of Figure 1. Are Experiment Setup, Preprocessing, Model Setup class names or verbs? Is Hyperparameter a class name or terminology from ML? If it is the latter, why is it italicized, and why if it is the former, then why is it not mentioned in Line 91? Similarly, the text used in the various bubbles in Figure 1 itself could benefit from such formatting (is Run Environment a class name or a simply a description?).

As it stands, the manuscript's treatment of different terminologies is too confusing for the reader to precisely understand what each term stands for. The manuscript would also benefit from a sentence or two stating the typographical choices made by the authors."

We have to agree that our text styling was not clearly chosen and, further, was not consistently implemented throughout the manuscript. Therefore, we thank Reviewer #1 for this comment and the proposed solution. We have agreed on the following. *Frameworks*, including *MLAir*, are italicised. For `code` elements such as class names and variables, the typewriter font is used. All other expressions, e.g. those that describe a class but do not explicitly name it, will not be highlighted in the text at all.

We have updated all corresponding text passages according to the convention described above.

We also have added a short statement at the end of the introduction to easily understand the styles' meaning.

3. "Section 2.1 would benefit from a definition of the main components – Run Environment, Data Handler, Model Class – what they do, and how they fit into the design of MLAir. Currently, only Run Environment is described, and Data Handler has a fairly confusing description (Line 93: Data Handler is responsible for an accurate use of the data).

   I strongly recommend rewriting Section 2.1 with the reader in mind, to ensure that readers are systematically introduced to the concepts and/or stages underlying MLAir and the corresponding software components that implement these concepts and/or stages."

   Based on the suggestions of both Referees, we have reordered chapters 2 and 3 to make MLAir easier to understand. To do this, we have integrated the entire section 3 into chapter 2. Sections 2.2 to 2.4, on the other hand, have been moved to chapter 3. In the new chapter 2, the programming language is now discussed first (previously 3.1), then the general design is briefly explained (previously 2.1) and then the concepts of run modules, model class and data handler are explained (previously 3.2 to 3.4). In the newly created chapter 3, the order corresponds exactly to the former sections 2.2 to 2.4 .

4. "To ensure the a reader with no background in ML is able to relate to the design of MLAir, a brief introduction to a typical ML workflow (test, train, validation, epoch, hyperparameter, etc.,) would be really helpful in Section 2.1."

   We have added two paragraphs explaining ML in short and how a ML workflow could look like in the beginning of section 2.

5. "In Section 2.2, window_history_size is not defined anywhere. Therefore, what it does and how it might impact network architecture (Line 128–129) is not clear."

   We have adapted the manuscript in two places regarding this. First, we have referenced more precisely in l. 122 what the parameter window_history_size stands for.

   "Therefore, we need to adjust the parameter `window_history_size` for the

former and `stations` for the latter in the run call."
We have also added the following sentences to l. 129 in order to answer the question of how exactly the architecture is influenced.
"This is made possible since the model class in MLAir queries the shape of the input variables and adapts the architecture of the input layer accordingly. Naturally, this procedure does not make perfect sense for every model, as it only affects the first layer of the model. In case the shape of the input data changes to a large extent, it is advisable to adapt the entire model as well."

6. "Line 142: The term epoch is used without defining it anywhere. Definitions and motivations for such terms could be added to the brief description of a typical ML workflow that was mentioned previously."
We have added a short explanation for the term epoch in the new short introduction to ML.

7. "Line 220: If the skill score is defined as a ratio of a metric like MSE, how does one obtain positive and negative skill?"
We have expressed ourselves in a misleading way at this point. The skill score depends on the ratio but is not directly defined as the ratio. We have now split this information into two sentences and described it more precisely.
"For the comparison, we use a skill score $S$, which is naturally defined as the performance of a new forecast compared to a competitive reference with respect to a statistical metric (Murphy and Daan, 1985). Applying the mean squared error as the statistical metric, such a skill score $S$ reduces to unity minus the ratio of the error of the forecast to the reference."

8. "Line 229: The process of bootstrapped predictions could be explained a little more in detail for completeness. "the time series of each individual input variable is resampled n times (with replacement) and then fed to the trained network." was not sufficient for me (at least) to understand what was being attempted and

achieved."

We have added some information on how bootstrapping is done and what can be gained from it. By resampling a single input variable, its temporal information is disturbed, but the general frequency distribution is preserved. The second property is important because it ensures that the model is provided only with values from a known range and does not extrapolate out-of-sample. If the skill score is now calculated in comparison to the original prediction, it is to be expected that it will be negative because the information of the sampled variable has been lost. Conversely, the greater this drop, the stronger is the impact of this input variable on the prediction:

"In addition to the statistical model evaluation, *MLAir* also allows to assess the importance of individual input variables through bootstrapping of individual input variables. For this, the time series of each individual input variable is resampled $n$ times (with replacement) and then fed to the trained network. By resampling a single input variable, its temporal information is disturbed, but the general frequency distribution is preserved. The latter is important because it ensures that the model is provided only with values from a known range and does not extrapolate out-of-sample. Afterwards, the skill scores of the bootstrapped predictions are calculated using the original forecast as reference. If an input variable is important to achieve a good model forecast, it will thus show up with a large negative skill score in the bootstrap skill score plot (Fig. 13). Input variables that show an overly negative skill score during bootstrapping have a stronger influence on the prediction than input variables with a small negative skill score. In case the bootstrapped skill score even reaches the positive value domain, this could be an indication that the examined variable has no influence on the prediction at all. The result of this approach applied to all input variables is presented in `PlotBootstrapSkillScore` (Fig. 13). A more detailed description of this approach is given in Kleinert et al. (2021)."

9. "Figure 12: I might have missed something, but it is unclear what AI, BI, CI, CA-SEI etc., are. Furthermore, it is unclear what "terms" and "contributing terms" (Line 226) are."
We have slightly adjusted the sentence in l.226 and included a brief description on the climatological skill scores in the caption of Fig. 12.
"... and summarized as a full-detail box-and-whiskers plot over all stations and forecasts with all contributing terms (Fig. 12), and as simplified version showing the skill score only (not shown)." (l.226)
"Climatological skill scores (CASE I to IV) and related terms of the decomposition as proposed in Murphy (1988). Skill scores and terms are shown separately for all forecast steps (dark to light blue). In brief, CASE I to IV describe a comparison with climatological reference values evaluated on the test data. CASE I is the comparison of the forecast with a single mean value formed on the training and validation data and CASE II with the (multi-value) monthly mean. The climatological references for CASE III and IV are, analogous to CASE I and II, the single and the multi-value mean, however, on the test data. CASE I to IV are calculated from the terms AI to CIV. For more detailed explanations of the cases, we refer to Murphy (1988)." (Fig. 12)

10. "Lines 417-421: The line numbers in Figure 14 that are referred to here don't seem to correspond to their respective descriptions. I hope the authors verify the same for other such figures as well."
Lines numbers have been updated.

11. "Line 440: A more detailed description of how the imbalance problem is addressed would benefit the reader who wishes to use this feature."
The explanation of the imbalance problem was expanded by describing the procedure in more detail.
"This can be applied during training by using the `extreme_values` parameter, which defines a threshold value at which a value is considered extreme. Training

samples with target values that exceed this limit are then used a second time in each epoch. It is also possible to enter more than one value for the parameter. In this case, samples with values that exceed several limits are duplicated according to the number of limits exceeded.

Minor Comments

1. "Line 58: It also allows to deploy -> It also allows deploying"
   Corrected

2. "Line 59: use of GPUs . I think it is proper to acknowledge that usage of GPUs is due to the underlying tensorflow library"
   We added proposed acknowledgement
   "It also allows deploying typical optimization techniques in ML workflows, and offers further technical features like the use of graphics processing units (GPU) due to the underlying ML library."

3. "Line 60: Concurrent to a simple usage with low barriers for ML-callow scientists,: Not sure what is meant by this sentence."
   We have rephrased the sentence.
   "*MLAir* is suitable for ML beginners by its simple usage, but also offers high customization potential for advanced ML users and can therefore be employed in real-world applications."

4. "Line 84: as many customization -> To facilitate customization(?)"
   We have rephrased the beginning of the corresponding sentence to be more clear.
   "In order to enable a wide range of adaptations but also to support the users sufficiently, MLAir had to be designed as an end-to-end workflow comprising all required steps of the time series forecasting task."

[Figure]

5. "Line 83: Why is Workflow capitalized?"
   We had originally capitalized all nouns. We have now lowered all nouns in headings.

6. "Figure 1: exemplary: exemplary usually means commendable. I don't think that is what the authors meant."
   We have changed the beginning to
   "An exemplary example implementation of a little model using ..." (l.417)

7. "Figure 6: exemplary measurement stations: Same as above. This applies to all other places where exemplary has been used."
   Done

8. "Line 110: daily aggregated: does this mean daily mean? daily max? It is unclear how the aggregation is done."
   We have added the reference to a table with more details on the aggregation because it depends on the variable.
   "In the default configuration, 21-year time series of nine variables from five stations are retrieved with a daily aggregated resolution (see Table 3 for details on aggregation)." (l.109)

9. "Line 168: Beside the file, which contain the model -> Besides the file, which contains the model"
   Done

10. "Line 181: intended to add own graphics in MLAir -> intended to add custom graphics in MLAir"
    Done

11. "Line 185: major waters -> major water bodies"
    Done

12. "Line 191: are meant to give an insight into -> are meant to provide insight into"
Done

13. "Line 195: each month separately as box-and-whisker -> each month separately as a box-and-whisker diagram"
Done

14. "Line 205: marginal distribution is shown as histogram (lite grey) -> marginal distribution is shown as a histogram (light grey)"
Done

15. "Line 242: independent on the OS -> independent of the OS"
Done

16. "Line 350: Since the spatial dependency of two distinct stations may variegate related ->Since the spatial dependency of two distinct stations may vary(?) related"
Yes, replaced by vary

17. "Line 433: how to implement a ML model -> how to implement an ML model"
Done, also updated in l.4 and l.6.

18. "Line 436: is to assume an independent and identically distribution and therefore augment and randomly shuffle data to produce a larger number of input samples with a broader variety. -> is to assume independent and identically distributed data and therefore augment and randomly shuffle the data to produce a larger number of input samples with a broader variety. Consider rewriting this sentence to enhance readability."
We have split and rephrased the long sentence into two parts for better understanding.
"A popular technique in ML, especially in the image recognition field, is to augment and randomly shuffle data to produce a larger number of input samples with

a broader variety. This method requires independent and identically distributed data."

19. "Line 442: To address this issue, MLAir allows to place more -> To address this issue, MLAir allows placing more"
    Done

**3   Answer to Anonymous Referee #2**

"Through this manuscript, a new library specialized in Neural Networks for air quality forecasting is presented. In general terms, it is well written, the objectives are clear, and main features are correctly established.

Given that Neural Networks are gaining importance and widening their public, it is of high interest developing new tools that facilitate the intersection between this concrete field and any other area. In that sense, this work is an example of transparency and reproducibility, especially important in the Artificial Intelligence domain."

Comments

1. "The title is at some degree confusing. Using the term "Machine Learning" might lead to confusion, as the framework is solely based on deep learning."
   We prefer to keep the present title as MLAir can in principle be extended to other machine learning methods. However, addressing also comments by reviewer #1, we have added text in the abstract, the beginning of section 2 and in the limitations chapter to point out that the framework currently supports only shallow and deep neural networks.

2. "It feels like section 2 and 3 could be exchanged: before explaining how the framework is used, it might be interesting to know how the framework works. Also, the description of the framework is quite scarce and for a non-computer scientist reader it could be hard to follow. "
Following this comment, we have reordered chapters 2 and 3 to make MLAir easier to understand. To do this, we have integrated the entire section 3 into chapter 2. Sections 2.2 to 2.4, on the other hand, have been moved to chapter 3. In the new chapter 2, the programming language is now discussed first (previously 3.1), then the general design is briefly explained (previously 2.1) and then the concepts of run modules, model class and data handler are explained (previously 3.2 to 3.4). In the newly created chapter 3, the order corresponds exactly to the former sections 2.2 to 2.4 .

3. "Given that the manuscript aims to improve synergies between air quality experts/ practitioners and deep learning methodologies, some kind of introduction to deep learning main points is recommendable."
We added a brief general introduction to machine learning and the typical ML workflow to section 2. A more comprehensive introduction to deep learning would clearly go beyond the scope of this paper and there is ample good learning material available on the internet.

4. "Some terms are not described before being used. Again, as in Section 2 the functioning and experimentation procedure is presented without any previous explanation of the general framework, sometimes it is confusing."
This issue has been resolved by the restructuring related to comment 2.

5. "Grammar and typo errors should be carefully checked throughout the entire manuscript."
We have double-checked the manuscript for typos and grammar errors, but as non-native speakers we are aware that our use of the English language is not

perfect.

**4 Additional Changes**

1. We have changed "German Umweltbundesamt (UBA)" in l.383 to "German Environment Agency (Umweltbundesamt, UBA)"

2. We have updated the reference Kleinert (2020) -> Kleinert (2021) because it has now been published.

3. We have updated the publication year of Schultz et al. (2020) -> Schultz et al. (2021), because the publication release is scheduled for February 2021.

4. We have updated the description on the paper's structure (l.67)

5. We have added a reference for the hierarchical data format in l.169: Koranne (2011)

6. We have also added a reference for the json format in l.192: ISO Central Secretary (2017)

7. Another reference has been added for jupyter notebooks in the caption of Fig. 2: Kluyver et al. (2016)

8. We now have explicitly stated the name of each plot class in the plot description.

9. We have added a reference for Markdown and LaTex in l. 160: Gruber (2004) and LaTeX Project (2005)

10. We have updated Fig. 1 to be consistent with the highlighting

11. We have added a reference entry for the source code: Leufen et al. (2020)

12. We have corrected grammar or typo errors in l. 143, 243, 251, 279, 288, 312

**5 References**

- Gruber, J.: Markdown, https://daringfireball.net/projects/markdown/license, (accessed on January 07, 2021), 2004.

- ISO Central Secretary: Information technology — The JSON data interchange syntax, Standard ISO/IEC 21778:2017, International Organization for Standardization, Geneva, CH, https://www.iso.org/standard/71616.html, 2017

- Kleinert, F., Leufen, L. H., and Schultz, M. G.: IntelliO3-ts v1.0: a neural network approach to predict near-surface ozone concentrations in Germany, Geoscientific Model Development, 14, 1—25, https://doi.org/10.5194/gmd-14-1-2021, 2021

- Kluyver, T., Ragan-Kelley, B., Pérez, F., Granger, B., Bussonnier, M., Frederic, J., Kelley, K., Hamrick, J., Grout, J., Corlay, S., Ivanov, P., 630 Avila, D., Abdalla, S., Willing, C., and development team, J.: Jupyter Notebooks - a publishing format for reproducible computational workflows, in: Positioning and Power in Academic Publishing: Players, Agents and Agendas, edited by Loizides, F. and Scmidt, B., pp. 87–90, IOS Press, Netherlands, https://eprints.soton.ac.uk/403913/, 2016.

- Koranne, S.: Hierarchical data format 5: HDF5, in: Handbook of Open Source Tools, pp. 191–200, Springer, HDF5 is maintained by The HDF Group, http://www.hdfgroup.org/HDF5, 2011

- LaTeX Project: LaTeX, https://www.latex-project.org/, (accessed on January 07, 2021), 2005

- Leufen, L. H., Kleinert, F., and Schultz, M. G.: MLAir (v1.0.0) - a tool to enable fast and flexible machine learning on air data time series - Source Code,

EUDAT Collaborative Data Infrastructure, https://doi.org/http://doi.org/10.34730/fcc6b509d5394dad8cfdfc6e9fff2bec, 2020

- Murphy, A. H.: Skill Scores Based on the Mean Square Error and Their Relationships to the Correlation Coefficient, Monthly Weather 620 Review, 116, 2417–2424, https://doi.org/10.1175/1520-0493(1988)116<2417:SSBOTM>2.0.CO;2, 1988.

- Murphy, A. H. and Daan, H.: Forecast evaluation, in: Probability, statistics, and decision making in the atmospheric sciences, edited by Murphy, A. H. and Katz, R. W., pp. 379—-437, Westview Press, Boulder, USA, 1985

---

## Referee Report (RR1)

**Review of Revision of MLAir (v1.0) - a tool to enable fast and flexible machine learning on air data time series by Leufen et al.**

I find that the changes made by the authors in this revision address the concerns I raised in my previous review. There continue to be a few issues with language, some of which I have highlighted below. I recommend that the authors pass through the manuscript once more with an eye towards readability.

Once these minor changes have been made, this manuscript should be ready to be accepted at GMD.

**1 Minor Comments**

1. Line 86: This is an iterative procedure, a single iteration is called epoch → This optimisation is an iterative procedure and each iteration is called an epoch

2. Line 104: As underlying coding language python (Python Software Foundation, 2018, release 3.6.8) was used for two major reasons → python (Python Software Foundation, 2018, release 3.6.8) was used as the underlying coding language python for two major reasons

3. Line 109: Secondly, python is currently the language in the ML community *to* Secondly, python is currently the language of choice in the ML community

4. Line 122: and introduces labels in form of dimensions *to* and introduces labels in the form of dimensions

5. Line 151: all local paths for the experiment itself but also for data are created *to* all local paths for the experiment and data are created

6. Line 161: Right after, the actual training starts *to* The actual training starts subsequently

7. Line 162: If performance improved compared to previous cycles *to* If performance improves as compared to previous cycles

8. Line 163: In this way, the final model is the best training model according to validation performance. *to* Needs to be reworded

9. Line 186: data retrieval, preparation and provision of a single data origin *to* Not sure what "single data origin" means here

10. Line 224: we expand the number of precedent time steps *to* we increase the number of observations? Not entire sure what is the best way to reword this. But precedent time steps is awkward.

11. Line 261-62: Even if the batch data could be used further, they serve rather as auxiliary files. *to* Could you elaborate this further? I'm unclear what this means.

12. Line 353: MLAir offers a high number *to* MLAir offers a large number

13. Line 400: feature the parameter upsamling *to* feature the parameter up-sampling

14. Line 433: temporal resolution of the data is set with sampling: What datatype is required here? int/float/datetime?

---

## Author Response (AR2)

**Authors response to referee comments: MLAir (v1.0) - a tool to enable fast and flexible machine learning on air data time series**

Lukas H. Leufen[1,2], Felix Kleinert[1,2], and Martin G. Schultz[1]

[1]Research Centre Jülich, Jülich Supercomputing Centre, Germany
[2]University of Bonn, Institute of Geosciences, Germany

**Correspondence:** LH Leufen (l.leufen@fz-juelich.de)

**1 General statement**

We have incorporated Reviewer #1's comments into the text and also went through the entire manuscript again. Now, we hope for publication. Line numbers given in this document always refer to the revised version *gmd-2020-332-manuscript-version3.pdf*. All changes are highlighted in blue in the track-changes file. There is no distinction between changes made to answer a reviewer comment or by ourselves in this version.

**2 Answer to Anonymous Referee #1**

"I find that the changes made by the authors in this revision address the concerns I raised in my previous review. There continue to be a few issues with language, some of which I have highlighted below. I recommend that the authors pass through the manuscript once more with an eye towards readability. Once these minor changes have been made, this manuscript should be ready to be accepted at GMD."

**Minor Comments**

1. "Line 86: This is an iterative procedure, a single iteration is called epoch -> This optimisation is an iterative procedure and each iteration is called an epoch"

   Done

2. "Line 104: As underlying coding language python (Python Software Foundation, 2018, release 3.6.8) was used for two major reasons -> python (Python Software Foundation, 2018, release 3.6.8) was used as the underlying coding language python for two major reasons"

   Done

3. "Line 109: Secondly, python is currently the language in the ML community *to* Secondly, python is currently the language of choice in the ML community"

   Done

4. "Line 122: and introduces labels in form of dimensions *to* and introduces labels in the form of dimensions"

   Done

5. "Line 151: all local paths for the experiment itself but also for data are created *to* all local paths for the experiment and data are created"

   Done

6. "Line 161: Right after, the actual training starts *to* The actual training starts subsequently"

   Done

7. "Line 162: If performance improved compared to previous cycles *to* If performance improves as compared to previous cycles"

   Done

8. "Line 163: In this way, the final model is the best training model according to validation performance. *to* Needs to be reworded"

   We have rewritten the sentence.

9. "Line 186: data retrieval, preparation and provision of a single data origin *to* Not sure what "single data origin" means here"

   We have made the sentence more generic, as the statement "single data origin" was only misleading at this point.

10. "Line 224: we expand the number of precedent time steps *to* we increase the number of observations? Not entire sure what is the best way to reword this. But precedent time steps is awkward."

    The sentence is rephrased now and the word "precedent" is replaced by "previous".

11. "Line 261-62: Even if the batch data could be used further, they serve rather as auxiliary files. *to* Could you elaborate this further? I'm unclear what this means."

    We have deleted this sentence from the manuscript because we ourselves are not aware of any meaningful use for it and this note is therefore dispensable.

12. "Line 353: MLAir offers a high number *to* MLAir offers a large number"

    Done

13. "Line 400: feature the parameter upsamling *to* feature the parameter upsampling"

    Done

14. "Line 433: temporal resolution of the data is set with sampling: What datatype is required here? int/float/datetime?"

    We have added the datatype.

**3 Additional Changes**

In addition to the minor comments of Referee #1, we followed the recommendation and went through the entire manuscript again intensively. We have worked on improving the language and clarifying statements in several places. We do not list these changes in detail here, but refer to the track-changes file.